# Association of Blood Metabolomics Biomarkers with Brain Metabolites and Patient-Reported Outcomes as a New Approach in Individualized Diagnosis of Schizophrenia

**DOI:** 10.3390/ijms25042294

**Published:** 2024-02-15

**Authors:** Wirginia Krzyściak, Beata Bystrowska, Paulina Karcz, Robert Chrzan, Amira Bryll, Aleksander Turek, Paulina Mazur, Natalia Śmierciak, Marta Szwajca, Paulina Donicz, Katarzyna Furman, Fabio Pilato, Tamas Kozicz, Tadeusz Popiela, Maciej Pilecki

**Affiliations:** 1Department of Medical Diagnostics, Jagiellonian University Medical College, Faculty of Pharmacy, 30-688 Krakow, Poland; paulina.pater@uj.edu.pl; 2Department of Biochemical Toxicology, Jagiellonian University Medical College, Faculty of Pharmacy, 30-688 Krakow, Poland; beata.bystrowska@uj.edu.pl; 3Department of Electroradiology, Jagiellonian University Medical College, Faculty of Health Sciences, 31-126 Krakow, Poland; pkarcz@su.krakow.pl; 4Department of Radiology, Jagiellonian University Medical College, Faculty of Medicine, 31-503 Krakow, Poland; robert.chrzan@uj.edu.pl (R.C.); amira.bryll@uj.edu.pl (A.B.); tadeusz.popiela@uj.edu.pl (T.P.); 5Department of Child and Adolescent Psychiatry and Psychotherapy, Faculty of Medicine, Jagiellonian University Medical College, 31-501 Krakow, Poland; alek.turek@doctoral.uj.edu.pl (A.T.); natalia.smierciak@uj.edu.pl (N.Ś.); marta.szwajca@uj.edu.pl (M.S.); paulina.donicz@uj.edu.pl (P.D.); katarzyna.furman@gmail.com (K.F.); maciej.pilecki@uj.edu.pl (M.P.); 6Neurology, Neurophysiology and Neurobiology Unit, Department of Medicine, Università Campus Bio-Medico di Roma, 00128 Rome, Italy; f.pilato@unicampus.it; 7Department of Clinical Genomics, Center for Individualized Medicine, Mayo Clinic, Rochester, MN 55905, USA; kozicz.tamas@mayo.edu

**Keywords:** metabolomics, personalized approach, lactate, glutamate, cortisol, patient-reported outcomes, schizophrenia, magnetic resonance spectroscopy, psychosis

## Abstract

Given its polygenic nature, there is a need for a personalized approach to schizophrenia. The aim of the study was to select laboratory biomarkers from blood, brain imaging, and clinical assessment, with an emphasis on patients’ self-report questionnaires. Metabolomics studies of serum samples from 51 patients and 45 healthy volunteers, based on the liquid chromatography-electrospray ionization-mass spectrometry (LC-ESI-MS/MS), led to the identification of 3 biochemical indicators (cortisol, glutamate, lactate) of schizophrenia. These metabolites were sequentially correlated with laboratory tests results, imaging results, and clinical assessment outcomes, including patient self-report outcomes. The hierarchical cluster analysis on the principal components (HCPC) was performed to identify the most homogeneous clinical groups. Significant correlations were noted between blood lactates and 11 clinical and 10 neuroimaging parameters. The increase in lactate and cortisol were significantly associated with a decrease in immunological parameters, especially with the level of reactive lymphocytes. The strongest correlations with the level of blood lactate and cortisol were demonstrated by brain glutamate, N-acetylaspartate and the concentrations of glutamate and glutamine, creatine and phosphocreatine in the prefrontal cortex. Metabolomics studies and the search for associations with brain parameters and self-reported outcomes may provide new diagnostic evidence to specific schizophrenia phenotypes.

## 1. Introduction

Despite significant healthcare system reforms and advances in treatment, schizophrenia remains a debilitating severe mental disorder. Its estimated economic burden in the United States of America alone has doubled, reaching a staggering $343.2 billion in 2019 [1]. The indirect costs of this disorder result from frequent complications, including an increased number of suicides, substance abuse, and the ineffectiveness of applied antipsychotic treatment [2,3]. The current state of knowledge indicates the complex nature of the disorder, revealing gaps in the etiopathogenesis and diagnosis of schizophrenia [4]. These data justify the need for advancement in research including personalized approach to the disorder through the individualization of diagnostics and effective treatment tailored to the needs of the patient. The aim of this study was to select a set of blood metabolomic biochemical indicators and assess their relationship with brain metabolic parameters and clinical evaluation, with a particular emphasis on self-reported results filled out by patients with schizophrenia. 

Over the past decade, substantial evidence has emerged indicating structural and metabolic changes in the blood and brain of individuals with schizophrenia. Technological advancements and the complex nature of the disorder shed light on the intricate interactions between the host and its immune system, environmental influences, and a myriad of other yet unknown factors responsible for the pathophysiology of schizophrenia. Diagnosing the disorder often relies heavily on the psychiatrist’s conversation with the patient, which, in cases of advanced schizophrenia or its initial symptoms, can pose significant diagnostic challenges due to communication difficulties with the patient or the lack of objective, measurable biomarkers of disorder progression. Beyond targeted disorder diagnostics, there is also a need for an individualized approach to the patient through effective treatment and a more efficient response to therapy. Addressing existing gaps necessitates the development of a diagnostic panel of objective, measurable schizophrenia indicators

Ion and mass mobility spectrometry is one of the newer techniques for the analysis of complex compounds [5,6,7,8]. The roots of ion mobility lie in the discovery of Faraday’s electromagnetic induction phenomenon and the general theory of electrolysis in the early 19th century [9]. Rutherford and Thomson applied this knowledge to investigate the conduction of electricity in gases in 1896 [10]. In turn, the 1910s brought an increase in interest in the ionization of gases, the mobility of ions in air and gas, and the influence of electric fields on the mobility of gas ions [11,12]. However, these discoveries appeared only in the 1970s, when the first available ion mobility spectrometer (IMS) was constructed, based on the analysis of ionized molecules separated in the gas phase.

IMS is an analytical method that has been developing rapidly, especially in recent years. Separations using IMS are combined with gas and liquid chromatography. The IMS-MS method is a combination of IMS and mass spectrometry (MS) [13]. Dole used IMS in the development process of electrospray ionization (ESI) [14]. Metabolomics has recently become increasingly popular in diagnostic and biotechnological research. Metabolomics research are limited by problems in identification of metabolites, due to, among others, metabolic structural diversity and matrix interference effects [15,16]. The combination of IMS with other techniques allows to analysis of various biological materials with high resolving power, high sample throughput and with high sensitivity, additionally it also enables the determination of the structure of chemical compounds and the differentiation of isomers. These methods are therefore used in metabolomic research [13,14].

The liquid chromatography-electrospray ionization-mass spectrometry (LC-ESI-MS/MS) is a rapidly evolving analytical technique and one of the most sensitive methods in metabolomic research, providing structural and quantitative information in the identification of peptides, lipids, carbohydrates, and other metabolites in biological samples [5]. Previous metabolomic studies of biological materials (blood, urine, saliva, hair) from patients with schizophrenia [17,18,19,20,21,22] highlight the role of metabolites as potential biomarkers for schizophrenia. Metabolomic analysis of biological samples from patients based on LC-ESI-MS/MS is providing an increasing number of new disease biomarkers [23]. However, due to the complex nature of schizophrenia, there is a lack of studies assessing the relationship between metabolites in biological samples from patients, brain metabolites, and clinical evaluations and self-reports from patients supported by reliable research techniques.

In line with the World Health Organization (WHO) policy aiming to “define a plan of action to improve mental health for the years 2013–2030” [24] and to determine reliable, objective, measurable biomarkers for the polygenic nature of schizophrenia, we conducted a metabolomic study of blood serum samples from patients with schizophrenia, seeking connections with the results of neuroimaging analyses using Magnetic resonance imaging (MRI) and Magnetic resonance spectroscopy (MRS), psychiatric and psychological questionnaire interviews, and self-reports submitted by patients with schizophrenia.

## 2. Results

### 2.1. Distributions of Clinical Parameters by Groups

Distributions of clinical parameters by groups with estimation of differences between groups are presented in Table 1.

From the data in Table 1 there are significant differences between the control and test groups within four variables (the last two represented the same parameter): glutamic acid [µg/mL], cortisol [ng/mL], lactates [µg/mL], lactates [mmol/L]. The test group was characterized by lower values in all of the mentioned variables. The largest effect size was shown for lactates (moderate effect), the differences between the groups for the other variables were characterized by a small effect size. The estimated optimal cutpoints with the metric performance of binary classification (test or control group) for parameters with significant difference between groups from the Table 1 are shown in Table 2.

The results in Table 2 show that the classification metrics for patient groups are rather average and consist mainly of a rather high number of false negatives based on individual characteristics (especially for glutamic acid). In the case of glutamic acid and lactate, the classification metrics derived from the calculated cutoffs had quite high sensitivity parameters and can be used in situations where identification of a positive class (schizophrenia) is critical. In contrast, these metrics performed poorly (at the level of guesswork) in situations where classification of a negative class (healthy person) is crucial. This was mainly due to the overlap of the distributions for the two groups and should therefore be considered comprehensive for the purpose of classification. Fairly average, but the most balanced results in terms of absence and presence of the studied trait compared to the other studied traits showed the factor cortisol concentration. All metrics were better than the guessing level. Using the results in Table 2, it was shown that a blood test result with a cortisol level of 131.19 [ng/mL] or less, glutamic acid of 1671.59 or less [µg/mL], and lactate of 112.15 [µg/mL] allowed the patient to be classified in the test group.

### 2.2. Correlation Analysis of Lactates, Glutamic Acid and Cortisol Concentrations and Questionnaire Results in the Test Group

Table 3 shows the results of the correlation analysis of lactates, glutamic acid and cortisol concentrations with the results of 14 tests.

An increase in lactate concentration was significantly associated with an increase in *gastrointestinal symptom* scores, GHQ-28, and the *avoidance dimension* of the ITQ questionnaire (the strongest association). In turn, the increase in glutamic acid concentration [µg/ml] showed positive correlations with the arousal of *uncontrolled hostility excitement*, the dimensions of the PANSS questionnaire, the GHQ-28 scores, the *re-experiencing trauma, avoidance*, and dimensions of the ITQ questionnaire (the largest correlations). Furthermore, the change in cortisol concentration showed negative correlations only with the physical neglect dimension of the CTQ questionnaire. Additionally, the attached Appendix A presents results of the correlation of glutamic acid concentration in the blood with the total score of the scale (T). These results are addition to our previous work [25] and indicate that patients with a low change in T scores have significantly higher concentration of glutamic acid in the blood than patients with high change in T scores.

### 2.3. Correlation Analysis of Lactates, Glutamic acid and Cortisol Concentrations and Selected Clinical and Brain Parameters

The results of correlation analysis between lactates [µg/mL], glutamic acid [µg/mL], cortisol [ng/mL], selected laboratory, clinical and brain parameters are shown in Table 4.

From the data in Table 4, there were significant correlations between lactates [µg/mL] and 11 laboratory parameters and 10 brain parameters. An increase in lactates was significantly associated with a decrease in NEUT [×10^3^/µL], Re-Lymph [×10^3^/µL], NEUT [%], Re-Lymph [%], glucose [mmol/L], and an increase in Lymph [%], HDL cholesterol [µmol/L], FT4 [pmol/L]. Among the clinical parameters, Re-Lymph [×10^3^/µL] showed the strongest association with lactates.

Only the anterior cingulate cortex (ACC) within brain parameters were found to exhibit significant correlations with lactates. Specifically, at a TE of 30 ms, positive correlations were found in the case of creatine concentration [×10^6^], Cr+PCr concentration [×10^6^], and Glu+Gln concentration [×10^6^]. For TE 144 ms parameters, significant positive correlations were observed for glutamate concentration, Glu/Cr+PCr concentration, N-Acetylaspartate (NAA) concentration (which exhibited the highest correlation among all brain parameters), N-Acetyaspartate + N-Acetylspartylglutamate concentration, Cr+PCr concentration, and Glu+Gln concentration.

An increase in glutamic acid [µg/mL] resulted in a significant decrease in NEUT [%], glucose [mmol/L] (the highest correlation). A positive correlation behind glutamic acid [µg/mL] was shown for EO [%] and the only significantly related posterior cingulate cortex (PCC) brain parameter, at a TE of 30 ms, glutamine Cr+PCr.

Cortisol concentration was negatively correlated with Re Lymph [×10^3^/µL] and GI [×10^3^/µL] and positively correlated with cholesterol HDL [µmol/L]. For the parameters of PCC in TE 30 ms, the same positive associations were shown with glutamine Cr+PCr, N-acetylaspartate concentration [×10^6^], Cr+PCr concentration [×10^6^] (the highest association), Glu+Gln concentration [×10^6^]. At PCC in TE 30 ms, negative correlations of cortisol with taurine concentration [×10^6^] and Cr+PCr concentration [×10^3^] of taurine were found. At PCC in TE 144 ms, the increase in cortisol was significantly correlated only with the increase in L-alanine concentration.

### 2.4. The Cluster Analysis of the Test Results in the Test Group

As a result of the clustering procedure for the factor map estimated for the four dimensions, the results of the 14 tests performed in the test group were assigned to three clusters. The location of the clusters in two-dimensional space (Dim1, Dim1), which explain 41.7% and 17.1% of the variance in the data, respectively, is shown in Figure 1.

The characteristics of the individuals’ profiles based on the obtained test results were shown in Table 5. The absence of a test in a cluster meant that there were no significant differences compared to the mean of the test group.

From the data in Table 5, patients belonging to the first cluster are characterized by high GAF test scores and low scores on tests for gastrointestinal symptoms, ITQ, CTQ, GHQ-28, STAI), CALGARY, BDI II, SAPS, SANS, and PANSS. The results of the tests CISS, ECR-RS, TEC PL) were not significantly different from the group average. In contrast, patients in second cluster had scores significantly above the group average on the tests SANS, PANSS, SAPS and low scores on the tests ECR-RS, CTQ, STAI, BDI II, ITQ, and GAF. Test scores for CALGRY, gastrointestinal symptoms, CISS, GHQ-28, and TEC PL were not significantly different from the group average. Finally, patients in the third cluster had high scores on the tests BDI II, ITQ, STAI, CTQ, GHQ 28, DBZ RZ, gastrointestinal symptoms, TEC PL, CALGARY, CISS. Regarding the tests PANSS, SANS, SAPS, GAF scores were not significantly different from the group average.

## 3. Discussion

### 3.1. Distributions of Clinical Parameters by Groups

In our research the results of blood cortisol, glutamic acid, and lactates concentration assessed by LC-MS method were significantly lower in patients with schizophrenia than in control group. Cortisol, the main hormone secreted in the hypothalamus-pituitary-adrenal axis (HPAA), is responsible for the response to internal and external stress factors. In course of schizophrenia and other mental disorders, the hypothalamus-pituitary-adrenal axis is dysregulated. This dysregulation is primarily change in basal cortisol secretion in response to stress factors [26,27]. Decreased cortisol concentration may result from impaired stress response or be caused by antipsychotic medications [28]. Van den Heuvel et al. analyzed the level of hair cortisol concentration by liquid chromatography tandem mass spectrometry in a group of patients with schizophrenia, obtaining a significantly lower concentration of this parameter compared to the control group [29]. The authors emphasize that schizophrenia may be associated with impaired functioning of the HPAA, and thus with a reduced cortisol secretion [29]. In their research, Seitz et al. observed lower cortisol concentration in the saliva of patients with the first episode of psychosis (FEP) compared to the control group. The authors indicate that obtained result may be related to the phase of FEP (subacute, acute) or type of taken medications [30]. Tobolska et al. observed lower basic cortisol concentration in the saliva of patients with schizophrenia treated with antipsychotic drugs in comprehension with control group [26]. In their work, Zhang et al. observed reduced blood cortisol concentration caused by the use of risperidone and haloperidol taken by patients with schizophrenia [31]. Among our patients who used risperidone monotherapy, cortisol levels were also among the lowest level, for example: 60.96 ng/mL. Similarly, after haloperidol, the cortisol value was 66.68 ng/mL. A similarly low cortisol value was recorded in a patient after aripiprazole, but during polytherapy with lamotrigine and sertraline. Likewise, Woldesenbent et al., in their study on the impact of antipsychotic treatment on the cortisol level in a group of patients with newly diagnosed schizophrenia, observed a significant decrease the initial blood cortisol concentration after treatment [32]. Glutamate, the anionic form of glutamic acid, is the main excitatory neurotransmitter in the brain, the homeostasis of which is regulated by glutamate transporters [33]. Decrease of glutamate concentration in schizophrenia may be the result of impaired function of glutamatergic neurons. Kim et al., over 40 years ago, observed lower concentration of glutamate in cerebrospinal fluid of patients with schizophrenia [34]. Changes in schizophrenia glutamatergic neurotransmission affects the symptoms of the disorder and may be regulated by antipsychotic treatment [35]. Palomino et al. tested glutamate concentration in plasma of patients with schizophrenia and bipolar disorder. The authors observed decreased concentration of glutamate during the first psychotic episode. These results may indicate impaired glutamate signaling already at the initial stage of the disorder [36]. In the study of Marsman et al. proton magnetic resonance spectroscopy (^1^H-MRS) was used to determine glutamate level in brain of schizophrenia patients. In medial frontal region glutamate level was decreased in schizophrenia patients, compared to control group. Additionally, the authors showed a faster decline in glutamate levels with age. The obtained results may indicate that impaired glutamate transport, dysfunction in glutamate receptor functioning or altered synaptic activity may be involved in the development of schizophrenia [37]. In our previous research, glutamate, a parameter with statistically significant correlation with the PANSS scale, was used to distinguish two endophenotypes of schizophrenia- Type I with higher glutamate level in brain and Type II with reduced glutamate level in brain, compared to the control group. Endophenotype of schizophrenia with reduced glutamate level is associated with a more severe course of the disorder, co-occurring untreatable negative symptoms and the need for more intensive treatment [25]. The change in lactate concentration in patients with schizophrenia may indicate mitochondrial dysfunction and a change in energy metabolism from aerobic to anaerobic [38]. The increase in glucose level and decrease of lactate concentration in cerebrospinal fluid of patients with paranoid schizophrenia compared to the control group were observed in their study by Huang at al. These results may indicate that the brain of schizophrenia patient uses lactate as an energy source instead of glucose [39]. Merritt’s et al. meta-analysis of 42 studies involving 1251 patients with schizophrenia suggests that reduced brain glucose levels may be due to antipsychotic treatment, and the higher level of brain glucose may be used as a marker of disorder severity [40]. Onozato et al. in their research observed significantly lower concentration of L-lactate in serum of patients with at-risk mental state compared to control group, and no differences of D-lactate concentrations between groups. The authors concluded that lower L-lactate concentration in serum of at-risk of mental state patients may precede the onset of psychosis [41]. In our research on acid stress in schizophrenia patients induced by increased level of brain lactate, we found association of significantly higher total score on PANSS scale with significantly lower level of lactate in brain, and in the second group of schizophrenia patients lower total PANSS score was correlated with significantly higher lactate level in brain. These results are connected with our results on schizophrenia endophenotypes [25,42].

### 3.2. Correlation Analysis of Lactates, Glutamic Acid and Cortisol Concentrations and Questionnaire Results in the Studied Group

Stress-related disorders, anxiety, PTSD, phobias, and panic are now a serious social problem. An increasing number of studies indicate that glutamatergic neurotransmission is involved in the response to stress factors and in the development of anxiety-related disorders [43,44]. Glutamate, an excitatory neurotransmitter in the brain, is involved in the development of states of anxiety and aggression through its action in the limbic system [45,46]. In our research we obtained positive correlation of the increase glutamic acid concentration in the blood [µg/mL] with the arousal of uncontrolled hostility excitement and with G1-G16 subscale of PANSS. PANSS is a scale of 30-items that allow the assessment of both positive and negative symptoms of schizophrenia. A score of PANSS is rated on the basis of clinical interview with the patients and clinically important information derived by family and other informants. The PANSS uncontrolled hostility excitement factor is one of the items on positive scale. In addition to assessing positive and negative symptoms, the PANSS scale also allows for the assessment of the patient’s general psychopathology [47,48]. Frank et al., in their research on traumatic brain injury (TBI), a condition associated with the development of mental disorders such as depression, anxiety, aggression, apathy and PTSD, showed that drugs acting on glutamatergic neurotransmission are effective in the treatment of TBI and disorders resulting from it. In their research, injured rats were treated with pyruvate as an example of blood glutamate scavengers (BGS). Pyruvate, when used as a drug for neuropsychiatric disorders, contributed to a significant reduction in blood glutamate levels, thereby leading to a lower brain-blood glutamate concentration gradient and lower brain glutamate concentration. Decreased level of glutamate in the brain were associated with reduced level of aggression, depression and anxiety, as well as improved social behavior in rats with TBI [49]. These results are consistent with the positive correlation of blood glutamate levels and uncontrolled hostility excitement found in this work. Results obtained in our previous work showed that glutamate level in ACC of schizophrenia patients was lower than in the control group. These findings may result from the impact on glutamatergic transmission of antipsychotic drugs taken by patients on a regular basis. In our study, patients with schizophrenia were characterized by a predominance of negative symptoms and a higher score on the Beck Depression Inventory compared to the control group, which may also explain the reduced level of glutamatergic transmission [50]. The Appendix A contains unpublished own results indicating a strong correlation of the glutamic acid level in the blood with the change in the total score of the scale (T). The results obtained in our research allowed us to distinguish a group of patients with low and high change in T score. Patients with low change in T score have significantly higher level of glutamic acid in blood compared to group of patients with high change in T score. These results are a phenomena and continuation of our published research on the division of schizophrenia patients into clusters according to the level of glutamate in the brain as mentioned above. Type II of schizophrenia endophenotypes is characterized by a significant association of reduced glutamate level in the brain with general, cognitive and negative symptoms and a low association of reduced glutamate level with positive symptoms. The positive symptoms of the disorder can be eliminated by using first generation antipsychotics in therapy, which do not affect the negative symptoms, therefore patients with low level of glutamate in the brain require more intensive therapy to improve the quality of life [25]. These changes may be caused by glutamatergic hyperexcitability, associated with an increased flow of excitatory neurotransmitters (higher glutamine to glutamate ratio), or improperly functioning neuroglial coupling. The impaired functioning of glutamatergic transmission may be influenced by abnormal levels of synaptic proteins, abnormal function of glial cells in the prefrontal cortex of the brain (PFC) and abnormal volumes of the cerebral ventricles [51,52]. In our work on establishing new cut-off ranges for thyroid stimulating hormone (TSH) as part of the individual process of diagnosing schizophrenia, we obtained a negative correlation between the TSH level and diffusion in the right frontal lobe (DEV). This may result from disorders in the right frontal lobe of the brain involving a disturbed flow of microelements from astrocytes to neurons. Disorders of neuronal connections in the frontal lobes are responsible for the occurrence of negative symptoms of schizophrenia [53].

In our research, we also obtained a positive correlation of the level of glutamic acid in blood with general psychopathology (G1–G16) subscale of PANSS. In the assessment of general psychopathology, among others, the following features are used: depression, anxiety, disorientation, poor attention, and active social avoidance. Major depressive disorder (MDD) is a mental disorder that affects personal and social life and may cause sleeping, eating and general health problems. This disorder, as with schizophrenia, may result from disorders of the glutamatergic system, which may be responsible not only for the development of the disorder, but is also the main therapeutic target [54]. Küçükibrahimoğlu et al., analyzed blood levels of glutamate, gamma-aminobutyric acid (GABA), and glutamine in a group of patients with major depression treated with fluoxetine or escitalopram. The obtained results allowed the authors to draw the conclusions that level of plasma glutamate was lower in patients with major depression compared to the healthy control group and level of glutamate decreased after 10 days of therapy with both antidepressant drugs. The authors also emphasize the influence of glutamate on the development of depression [55]. Similar conclusions were presented in the study of Altamura et al. Using linear discriminant analysis, the authors showed a high statistically significant difference between patients with major depression (untreated) and healthy control group, using glutamate plasma level as the differentiating variable. The obtained results indicate that the development of major depression is associated with disturbances in the level of glutamate in the blood [56]. Mitani et al. in their work on the usefulness of plasma amino acids in assessing the severity of depression obtained a positive correlation of plasma glutamate levels with the Hamilton Depression Rating Scale (HAM-D) scores. The authors emphasize the usefulness of blood glutamate in assessing the severity of depression [57]. Wojtas et al. in their research on the use of psychedelics in the treatment of depression, emphasize that the development of depression may be caused by dysregulated glutamatergic neurotransmission. The high level of glutamate in the blood decreases during treatment, and the severity of symptoms of depression positively correlates with the level of glutamate in the blood. The authors indicate the loss of glial cells as a probable mechanism for changes in glutamate concentration in the blood of depressed patients [58]. Stress is one of the factors that may influence the development and course of depression. Aghajani et al., in their study on change of level of plasma metabolites in mice with depression induced by stress, observed significantly reduced plasma level of glutamate in mice with depression compared to the control group [59]. Spontaneous decrease in the level of glutamate in the blood after brain injury in rats was observed in the work of Zlotnik et al. The authors explain the obtained results as a reaction to stress related to the trauma, whose purpose is to remove excessive amounts of glutamate from the brain. The obtained results indicate that the level of blood glutamate is regulated by hormones released in stressful situations [60].

We also obtained a positive correlation of the level of glutamic acid in the blood with the General Health Questionnaire-28 (GHQ-28), the most common questionnaire for detecting emotional anxiety and mental disorders [61,62]. Disturbances in glutamatergic signaling have been associated with the pathogenesis of, among others, such mental disorders as schizophrenia, bipolar disorder, anxiety and depression [63]. Plasma glutamate concentration is associated with the severity of depression [64]. Luykx et al. in their meta-analysis, draw attention to brain region-specific changes in depression that are dependent on glutamate level [65]. Therefore, an altered level of glutamate in the blood may be associated with the risk of developing psychotic disorders. Inflammation is also considered to be the cause of the development of mental disorders such as schizophrenia or bipolar disorder. The functioning of the glutamatergic system is related to the functioning of the immune system, and disorders in the functioning of both systems underlie the development of psychiatric disorders [66,67]. Higher level of blood glutamic acid is often associated with depression, and lower level of glutamic acid concentration in blood may be associated with the development of schizophrenia, as well as other mental disorders, such as bipolar disorder. These results indicate a relationship among disorders in glutamate signaling, level of glutamate in the blood and the development of mental illness [36,68]. Schiffer, in his work, indicates that disorders in the glutamatergic receptor system may change the risk of developing schizophrenia, bipolar disorder and depression, three mental disorders that have a huge impact on the economic situation [69].

The positive correlation between the occurrence of gastrointestinal symptoms and blood lactate levels supports our previous findings, which indicate that human gastrointestinal bacteria and their metabolites influence the appearance of gastrointestinal symptoms in people with schizophrenia [42]. Lactate from the intestinal microbiome promotes anxiety-like behavior through the lipid metabolism pathway [70], which is explained by the dysfunction of the gut-brain axis underlying these changes [71]. Factors that predict schizophrenia include inflammation, the appearance of which is the body’s natural response to infection [50,72]. However, chronic inflammation may induce a number of negative health effects, including deterioration of cognitive functions and persistent negative symptoms accompanying schizophrenia [73]. The accumulation of lactate in the blood is associated with a poor prognosis in patients with chronic disorders, which is a precursor of disorder exacerbation. This increase may be explained by impaired mitochondrial function, which has been reported even among patients with schizophrenia without treatment [74]. This leads to inhibition of oxidative phosphorylation of glucose, and thus changes glucose metabolism to anaerobic, increasing the level of lactate in the blood [75]. On the other hand, treatment with haloperidol and olanzapine, commonly used in the treatment of schizophrenia, was also associated with increased blood lactate concentration, which may serve as an early biomarker of extrapyramidal symptoms in patients treated chronically with antipsychotics [76].

ITQ is an important measure of the International Classification of Diseases (ICD-11) [77], that well describes trauma-related disorders, i.e., post-traumatic stress disorder (PTSD), the component of which concerns avoidance of traumatic experiences; showed a significant positive correlation with the level of lactate and glutamic acid in peripheral blood in a group of patients with schizophrenia. This may be related to intrauterine hypoxia, during which the cells’ demand for O_2_ becomes critical, and the O_2_ deficit in the blood (hypoxemia) may transform into hypoxia and ultimately generate a risk of organ damage, including cerebral edema in the fetus [78,79]. The increase in hydrogen (H^+^) concentration is reflected in a decrease in potential of hydrogen (pH), which may manifest itself in increased lactate levels not only in hypoxic brain tissue, but also in the blood due to the disruption of the blood–brain barrier (BBB). This may be a prognostic factor for the diagnosis of metabolic acidosis, which is commonly defined as pH < 7.00 combined with base deficiency ≥ 12.0 mmol/L in the blood of people exposed to traumatic experiences in fetal life, and subsequently in early childhood and adulthood [79,80]. Additionally, after severe multiple injuries in childhood or early adolescence, the so-called biphasic immune response with early hyperinflammation followed by immunosuppression is observed [80]. Many humoral and cellular factors are involved in this process. Depending on individual predispositions, the overall severity of the injury, and the type of injury, both reactions can have lasting consequences on the immune system. As we showed in our previous studies, components of the immune system, i.e., reactive lymphocytes, entered the predictive model of schizophrenia and turned out to be a predictive factor of the disease, independent of the used treatment [50].

In our results, we also obtained a positive correlation of the level of glutamic acid in the blood with re-experiencing trauma, avoidance, and PTSD dimensions of the ITQ questionnaire. In the study of the influence of BBB permeability on the relationship between glutamate concentration in brain and blood, Boyko et al. emphasize that a properly functioning BBB protects the brain against the neurotoxic and neurodegenerative effects of too high blood glutamate levels. In their research on the impact of impaired blood–brain barrier permeability in rats, the authors showed a strong correlation between glutamate levels in the blood, cerebrospinal fluid and in the brain. A properly functioning BBB protects the brain against the toxic effects of excess glutamate, which is the basis of TBI and other mental illnesses that are based on impaired BBB permeability [81]. Theories regarding stress and the mental disorders that develop as a result of it, including PTSD, focus on the prefrontal cortex (PFC) and the amygdala. Childhood trauma affects the connectivity of the amygdala and anterior midcingulate cortex and the amygdala and ventromedial prefrontal cortex, which indicates the long-term impact of traumatic experiences on glutamatergic transmission, and thus on the level of glutamate in the brain and in the blood [82]. Similar conclusions were made by Heim and Nemeroff. In their work, the authors emphasize that, according to the results of epidemiological studies, people exposed to traumatic experiences in childhood are more susceptible to developing behavioral disorders, anxiety disorders, hyperactivity disorder, PTSD and depression. Research indicates that stress affects the circuits of the central nervous system, causing their sensitization and thus increased sensitivity, which may be the basis for the development of anxiety and depression. Stress also affects neurotransmitter systems, including, to a large extent, glutamatergic neurotransmission [83].

Of all of the assessment clinical scales analyzed in the group of people with schizophrenia, physical neglect was negatively correlated with the level of cortisol in the blood in the presented results. In turn, in the study of Flory et al., in which a linear regression analysis was performed covering all five subscales (i.e., sexual abuse, physical violence, emotional abuse, physical neglect and emotional neglect), it was found that physical violence was associated with lower cortisol level in the blood. Whereas, physical neglect was associated with higher cortisol levels (beta = 0.36, *p* = 0.02) after taking into account other forms of violence. The results may therefore indicate that childhood trauma has long-lasting effects and different forms of trauma may have different biological effects [84]. In Mazer et al. study, cortisol measured in patients with borderline personality disorder (BPD) was significantly lower compared to bipolar disorder (BD) in the presence of emotional and physical neglect. A significant negative correlation between the intensity of hopelessness and the level of cortisol has become a feature differentiating BPD from early life stress (ELS) [85]. This may mean that the greater the exposure to stress in childhood, the greater the susceptibility to psychopathology, manifested by a higher concentration of cortisol in the blood. In the literature on the subject, these changes may indicate structural and functional brain abnormalities, which may be related to the limited flow of microelement particles from astrocytes to neurons, which may additionally lead to damage to the white matter and reduction of connections between neurons in this area of the brain [53]. The relationship between TSH and myoinositol in the ACC in our previous studies supports astroglial dysfunction in this brain area and the appearance of negative symptoms of schizophrenia that are persistent in treatment. These observations support the view that traumatic events experienced in childhood, such as physical neglect, can lead to mental illness in adulthood, including PTSD, which is associated with significant biological changes associated with cognitive impairment, functional decline hypothalamic-pituitary-adrenal (HPA) axis (hypocortisolism) and activation of the innate immune response (low-grade inflammation) [86]. Molecularly, these changes may also involve changes in fractional anisotropy (FA), especially in the cingulate cortex, which has functional consequences and is related to anxiety, a personality dimension that is believed to be a transdiagnostic risk factor for affective disorders [87]. The history of stress in early adolescence has become a factor differentiating BPD from BD, as well as assessing its relationship with clinical symptoms and specific neuroendocrine reactions in each of these diagnoses. Changes in blood cortisol in response to acute social stressors are a factor in the increased risk of developing the disease in adulthood [88].

### 3.3. Correlation Analysis of Lactates, Glutamic Acid and Cortisol Concentrations and Selected Clinical and Brain Parameters

Phosphocreatine (PCr) is a high-energy phosphate compound found in the large amounts in the brain and plays an important role in buffering cellular energy and energy transport, especially in tissues with high and variable energy demand, such as the brain. PCr measurement provides an information on energy processes taking place in the local microenvironment of brain tissues and allows for the assessment of mitochondrial function in vivo [89]. In turn, lactate is usually considered as a product of glycolysis in hypoxic conditions, although it may also be a by-product of aerobic glycolysis in cancer cells [90,91,92]. Research in recent years also indicates the role of lactate, especially l-lactate, in improving memory, increasing cerebral blood flow, improving cerebral energy metabolism and stimulating the nervous system to regenerate [42,93]. Testing the level of lactate in the blood and its strong relationship with the level of creatine and phosphocreatine in the prefrontal cortex may be an indicator of tissue oxidation and the level of energetic activation of neurons [94]. Increasing peripheral lactate levels allows for easier uptake by astrocytes and neurons, suggesting that lactate levels are higher in astrocytes and that lactate is transported from astrocytes to neurons via monocarboxylate transporters (MCTs) [95]. Energy metabolism changes from aerobic to glycolytic processes, providing high-energy compounds, such as adenosine triphosphate (ATP). This process depends on the activation of glycolysis in astrocytes through the uptake of increased extracellular glutamate with the influx of sodium (Na^+^) inside, thereby increasing the lactate concentration [96]. Moreover, a damaged blood-brain barrier allows the secretion of lactate from the blood into the brain, which may increase the accumulation of lactate in the brain [97]. The blood-brain barrier, apart from its important role in providing energy sources (glucose and lactate), also mediates in the supply of energy “buffer” (creatine). The obtained results confirm that the creatine/phosphocreatine shuttle system is based on a complex relationship between the blood-brain barrier, glial cells and neurons in order to maintain and ensure brain energy homeostasis [98]. Sundqvist et al., in their work on the preparation of a mathematical model for processing data regarding changes in brain metabolism obtained from MRS, checked how the concentrations of metabolites such as lactate, glutamate and aspartate change in response to visual stimuli. The authors observed, in response to single, double and sequence of visual stimuli, an increase in the level of lactate and glutamate and a decrease in the concentration of aspartate and glucose. Glucose in the described model is transformed in the process of glycolysis into pyruvate, hence the decrease in concentration. Pyruvate may be transformed into lactate in the metabolic process, hence the increase in lactate concentration. In the tricarboxylic acid cycle (TCA), pyruvate combines with oxaloacetic acid (OAA), which is converted to 2-oxoglutarate (OG) through several reactions. In the process of transamination, 2-oxoglutarate (OG) can form glutamate. Aspartate can be formed from oxaloacetic acid (OAA) via transamination. The described mechanisms may explain the changes in metabolite concentrations obtained by the authors of the study, as well as the correlation between lactate, N-acetylaspartate and glutamate level obtained in our study. In the same work, the authors also assessed, by measuring the amplitude of concentration changes for different signal lengths, how long the signal must last for the change in brain metabolite concentration to be measurable. The obtained results allowed the authors to conclude that as the signal length increases, the amplitude of the metabolic response increases. These results may explain the larger number of correlations obtained in our work for metabolite measurement results with longer echo time (144 ms) than with shorter echo time (30 ms) [99].

In our previous studies, changes in glutamate/glutamine transmission turned out to be crucial in patients with specific endophenotypes of schizophrenia, for whom negative symptoms turned out to be dominant [25]. In the study of Šagud et al., most negative symptoms in patients with schizophrenia showed a positive correlation with inflammation indicators [100]. Inflammation sensitizes the HPA axis, disrupting the negative feedback loop and further promoting chronic inflammatory responses [53]. Ongoing peripheral inflammation exacerbates central inflammation (neuroinflammation) through several mechanisms, including disruption of the blood-brain barrier, transport of immune cells, and consequently activation of glial cells [101]. This chronic exposure to stress induces an increased level of cortisol in the blood, which is an etiological factor for the risk of inflammation of the nervous system and dominant negative symptoms in the studied group of patients. Negative symptoms of schizophrenia are difficult to treat and constitute an unfavorable prognostic factor. The relationship between blood cortisol levels and changes in glutamine/glutamate transmission in the prefrontal cortex observed in our study may include, among others: changes in reward processing, mainly involving subcortical areas such as the basal ganglia [102]. It is a well-known fact that inflammation of the nervous system occurs in approximately 30% of patients with MDD and is associated with a more severe, chronic and treatment-resistant course. As our previous studies show, inflammation is not specific only to depression but has transdiagnostic effects visible in the studied group of patients with schizophrenia [103], suggesting a common etiological risk factor underlying psychopathology and metabolic disorders [50]. There are studies indicating the relationship between chronic stress, mild inflammation and depressive symptoms [104]. One contemporary etiological hypothesis for schizophrenia is that the interaction between host genotype, microbial infection, and chronic stress causes schizophrenia, which is mediated by neuroinflammation and gastrointestinal dysbiosis [42,105]. So far, this evidence has only been observational and does not necessarily establish a causal relationship. Under normal conditions, cortisol reduces inflammation. However, persistent chronic stress stimulates the HPA axis, causing excessive cortisol release, resulting in dysregulation of the glucocorticoid negative feedback loop due to glucocorticoid receptor resistance, leading to excessive activation of the immune system [106]. Dexamethasone does not inhibit adrenocorticotropic hormone (ACTH) and cortisol in a subgroup of patients with melancholic depression, suggesting an underlying dysfunction of the HPA axis in this group of patients. Moreover, the concentration of free cortisol in urine is approximately twice as high in patients with depression compared to patients without depression [107].

In our research, we also obtained a positive correlation of blood cortisol level with such brain metabolites as N-acetylaspartate (NAA), creatine, and glutamate. Anxiety disorders and major depression (MD) are some of the disorders whose pathophysiology is stress-dependent. These disorders cause changes in the level of brain metabolites, such as glutamate and N-acetylaspartate, in the stress-sensitive area of the brain, the ACC. In their research, Bonnekoh et al. examined the relationship between the level of hair cortisol and brain metabolites in response to long-term stress, and the obtained results indicated the influence of cortisol level on the level of N-acetylaspartate in ACC. The authors concluded that ACC metabolism is dependent on the HPA axis [108]. Similar results were obtained by Neylan et al., who examined the relationship between salivary cortisol and the level of N-acetylaspartate in the hippocampus of patients with PTSD. The authors obtained a positive correlation between cortisol concentration and NAA level, what indicates the trophic effect of cortisol in the hippocampus [109]. The level of brain metabolites in patients with PTSD was also examined by Schuff et al. The obtained results indicated a significant reduction in the levels of N-acetylaspartate and creatine measured in the hippocampus of patients with PTSD compared to the control group. The reduced level of NAA may result from disturbed neuronal processes and disturbed oxygen metabolism, on which the cerebral level of N-acetylaspartate depends. The reduced level of creatine in the hippocampus of PTSD patients may result from the replacement of nerve cells by glial cells [110]. Schubert et al., in their work on the influence of corticosteroids on changes in the hippocampus, changes in neuronal structure and disorders of cognitive functions, showed that rats with hypercortisolemia have significantly elevated levels of brain glutamate. This positive correlation of cortisol and glutamate level indicates an excitotoxic effect on the hippocampus induced by glutamate [111].

### 3.4. The Cluster Analysis of the Test Results in the Studied Group

Analyzing the results of the clusters in the group of people with schizophrenia, we noticed that the patients of the largest—the second cluster (n-23) obtained results significantly higher than the average for the entire group in the SANS, PANSS and SAPS tests, what may indicate a homogeneous clinical phenotype of the diagnosis of schizophrenia in terms of positive and negative symptoms, as well as, the severity of individual symptoms of the disease in this group of patients. Obtaining a relatively homogeneous group of 23 people with schizophrenia gathered in the second cluster is also confirmed by the low results of a comprehensive assessment of current mental, social and professional functioning measured on the GAF scale. These results, in turn, are related to low self-assessment results, which were performed by patients themselves using the Beck Depression Inventory BDI-II, which measures the severity of depression, or the Spielberger State-Trait Anxiety Inventory (STAI). Interestingly, self-report results regarding the experience of trauma (ITQ) and coping with stress also in childhood and adulthood were relatively low in this group, what may confirm the smaller impact of adverse childhood experiences on general functioning in adulthood in this group of patients. In the case of people from the second cluster, the diagnosis of schizophrenia was to a lesser extent associated with relational trauma or post-traumatic stress disorder, unlike people from the third cluster, for whom the level of trauma or stress was a dominant feature, as well as the severity of depression or the level of anxiety, which correlated with the dominant gastrointestinal symptoms in this group of patients. This may be explained by the results obtained by Glynn et al., whose research showed that PTSD occurred 36% more often in cohorts of patients with gastrointestinal symptoms compared to group of healthy people. Post-traumatic stress was found to exacerbate gastrointestinal symptoms among people with Crohn’s disease, with disease exacerbation being four times higher in people meeting criteria for probable PTSD compared to controls. Interestingly, post-traumatic stress symptoms affect approximately one in five people with gastrointestinal problems. Explanations for the above observations were sought in risk factors, including: poor social support, history of life’s adversities and poor experience of pain [112]. Moreover, there is increasing evidence of the co-occurrence of gut microbiome (GM) dysbiosis and microbial dysbiosis contributing to the development of neurocognitive disorders such as depression, anxiety, post-traumatic stress disorder and finally schizophrenia [113]. Another explanation may be cardiovascular diseases, especially hypertension, and immune-related disorders often reported and comorbid with PTSD. Post-traumatic stress disorder is associated with instability of the limbic system and changes in both the hypothalamic-pituitary-adrenal axis and the sympathetic-adrenal axis, what affect neuroendocrine and immune functions, which may characterize a group of patients with schizophrenia for whom the immunological concept of the disease seems to be crucial [50,114].

The evidence for an association between PTSD and gastrointestinal (GI) disorders is mixed, partly due to methodological differences between studies [115], although they may also be due to the phenotypic differences of the cohorts of studied patients with schizophrenia. Studies that combine gastrointestinal disorders into one group with people experiencing excessive stress in childhood seem justified. However, as our research shows, they should not obscure the rest of the clinical picture and the search for common correlates with other results of clinical assessment and self-report tools, the results of the functioning of hypothalamic-pituitary-adrenal axis and results of laboratory and imaging tests, which only when considered together make sense in showing the complex nature of the schizophrenia.

### 3.5. Limitations

The study has a number of limitations. The group of patients included in the study is characterized by different age of onset, duration of psychosis and its stage. Both male and female participants were included in the study. This may affect the dispersion of the results. Schizophrenia diagnosis is made based on the clinical picture. This means that the groups of patients included in the study are potentially heterogeneous in nature. The differential diagnosis of patients included in the research was not uniform and contained elements relating to the clinical picture. Therefore, patients were not subject to a uniform diagnostic procedure, including, for example, video-EEG examination and cerebrospinal fluid examination. However, none of the patients from the research and control groups showed features indicating the presence of anomalies in the MRI examination that would question the diagnosis of schizophrenia or indicate the need for further examination of the control group. The exclusion of mental and somatic disorders in the control group was based on patients’ declarations and interview without a full medical examination or additional tests. Based on basic additional tests, no disorders requiring further diagnostics were found in any person from the control group.

## 4. Materials and Methods

### 4.1. Participants

Recruitment and clinical assessment were carried out at the Clinical Department of Adult, Child and Adolescent Psychiatry of the University Hospital in Krakow. The study included people diagnosed with schizophrenia according to International Classification of Diseases (ICD-10) criteria, assessed by specialized and experienced psychiatrists. Using a structured clinical interview, the psychiatrist assessed the severity of schizophrenic symptoms using the Positive and Negative Syndrome Scale (PANSS) [25]. The age of the study participants ranged from 13 to 40 years. Adult participants gave informed consent to participate in the study. Participants under 18 years of age gave informed consent to participate in the study with the consent of the parents or legal guardians. The control group consisted of age- and gender-matched healthy volunteers who met specific demographic criteria. These participants did not have a diagnosis of acute psychosis, schizophrenia, or other mental disorders according to ICD-10 criteria.

Exclusion criteria from the study included court-ordered treatment, limited legal capacity, and intellectual disability. People suffering from severe cardiovascular diseases, diabetes, insulin resistance, metabolic syndrome, and a history of central nervous system diseases were also excluded from the study. People taking the following drugs in the last 3 months before the study were excluded: clozapine, people who had a change or modification of antipsychotic treatment recorded within 12 weeks before the study, use of anabolic-androgenic steroids and recent use (within 3 days before the study) of antibiotics, non-steroidal anti-inflammatory drugs, corticosteroids, vitamin supplements, antioxidants, probiotics, psychoactive or narcotic substances. People who were diagnosed with addiction to psychoactive substances according to ICD-10 or abuse of alcohol or psychoactive substances (except tobacco) within 3 months before the study were excluded from the study. Other exclusion criteria included intense affective symptoms, hyperactivity or psychomotor agitation, pregnancy, breastfeeding, or contraindications to MRS. The clinical diagnosis was made on the basis of a psychiatric examination and confirmed using PANSS [47]. The Scale for Assessment of Negative Symptoms (SANS) [116] was also used to assess negative symptoms. The severity of individual schizophrenic symptoms was assessed using a short Scale for Assessment of Positive Symptoms (SAPS) [117], which allows for reliable measurement and is sensitive to changes in the severity of symptoms. The clinical assessment also used the Calgary Depression Scale for Schizophrenia (CDSS), which is widely used to assess the severity of depression in schizophrenia [118]. Using the Global Assessment of Functioning (GAF) [119], a comprehensive assessment of the current mental, social, and vocational functioning of study participants was conducted [120]. The study participants self-assessed their well-being using the Beck Depression Inventory BDI-II, which measures the severity of depression in psychiatrically diagnosed patients [121], and also completed a shortened form of the Spielberger State-Trait Anxiety Inventory (STAI) [122,123]. Each participant also completed the Endler and Parker Coping Inventory for Stressful Situations (CISS) in the Polish adaptation by Szczepanik, Wrześniewski and Strelau [124] and the International Trauma Questionnaire (ITQ) [125] in order to assess coping with stress. Participants also completed retrospective self-assessments of negative and potentially traumatic experiences in childhood and adulthood using the Traumatic Experiences Checklist (TEC) [126], reflecting the total number of potentially traumatic and adverse events over the course of their lives. Study participants also completed the Childhood Trauma Questionnaire (CTQ) [127], which is designed to quantify the history of childhood trauma reported by young adults. The CTQ measures childhood trauma using five subscales: emotional abuse, physical abuse, sexual abuse, emotional neglect, and physical neglect. All participants completed the General Health Questionnaire-28 (GHQ-28) [61,62] as the most frequently used questionnaire to detect emotional distress and possible mental illness in the general population [128]. Gastrointestinal symptoms were assessed using the Gastrointestinal Symptom Rating Scale (GSRS) ([129]; Polish version [130]). All participants completed the State-Trait Anxiety Inventory (STAI) [122] and the self-report questionnaire: Experiences in Close Relationships-Revised-Short form (ECR-RS) in order to assess the individual attachment style on two dimensions of anxiety and avoidance experiences in close relationships ([131]; Polish version [132]).

### 4.2. Material

#### Characteristics of the Sample

The distributions of laboratory parameters (for the test group) are shown in Table 6 (for brain parameters distribution see previous analysis).

### 4.3. Methods

#### 4.3.1. Laboratory Routine Tests

In the morning, following an 8-h fasting period and overnight rest, on the day of blood collection, prior to medication intake (pertaining to chronic medications), comprehensive laboratory examinations were conducted on both patients and healthy volunteers. These examinations included a complete blood count, lipid profile (low-density lipoprotein, high-density lipoprotein, LDL and HDL; triglycerides, TG, and total cholesterol, TC), serum creatinine concentration, alanine aminotransferase (ALT) activity, inflammatory markers (high-sensitivity C-reactive protein, hsCRP), complement components C3 and C4, ionogram (potassium—K^+^, sodium—Na^+^, magnesium—Mg^2+^), glucose, insulin, uric acid, homeostasis model assessment of insulin resistance (HOMA-IR) index as well as thyroid function tests (free triiodothyronine, FT3, free thyroxine FT4, and thyroid-stimulating hormone TSH, released by the pituitary gland to stimulate thyroid production of FT3 and FT4), antibodies against thyroid peroxidase (anti-TPO), adrenal parameters assessment (dehydroepiandrosterone sulfate, DHEA-S), and serum ferritin levels.

Blood was collected in accordance with Good Laboratory Practice (GLP) and Good Clinical Practice (GCP) standards by qualified personnel. The blood was drawn into closed tubes containing a coagulation activator. The samples were gently mixed and kept at room temperature in an upright position for 30 min, after which they were centrifuged at 1300× *g* for 15 min. Following centrifugation, the serum was utilized for various assays. Hemolyzed and lipemic samples were discarded.

Routine analyses were conducted in the central laboratory of the University Hospital in Krakow using automated analyzers including the XN-9100 model (Sysmex Corporation, Kobe, Japan), biochemical analyzers Cobas 8000, PRO, and PRO/e801 (Roche Holding AG, Basel, Switzerland), as well as Siemens Behring Nephelometer BN II (Erlangen, Germany). The University Hospital Laboratory in Krakow is subject to daily internal quality control (Precicontrol ClinChem Multi 1, Precicontrol ClinChem Multi 2, Lyphochek Assayed Chemistry Control level 1, Lyphochek Assayed Chemistry Control level 2, Precicontrol Universal level 1, and Precicontrol Universal level 2 for biochemical parameters; Sysmex XN-Check at three levels) and systematic external quality control (conducted by the Central Laboratory Quality Assessment in Laboratory Diagnostics in Poland and the International Quality Assessment System by Randox), in accordance with established standards for medical diagnostic laboratories.

#### 4.3.2. Determination of Metabolites in Peripheral Blood

The Liquid Chromatography-Electrospray Ionization-Mass Spectrometry (LC-ESI-MS/MS) method was used to determine the concentration of alanine (ALA, ALA-d4), serine (SER, SER-d4), glutamic acid (GLUT, GLUT-C13), lactates (LA) and cortisol (COR). Acetic acid, propionic acid, butyric acid, isobutyric acid, valeric acid and isovaleric acid were determined using LCMS Short Chain Fatty Acids (SCFA) analysis. In this study Waters ACQUITY UPLC^®^H-Class LCMS chromatograph was used (Waters Corporation, Milford, MA, USA), equipped with a pump, thermostat, autosampler and a PDA detector (the purchase of the equipment was financed by the qLife program). Quantitative analysis was performed using a Xevo TQ-S^®^ quadrupole mass spectrometer (Waters Corporation, Milford, MA, USA) with electrospray ionization (ESI) in positive ionization mode (only in the case of lactates determination, negative ionization mode was used). Data collection was performed in selected multiple reaction monitoring (MRM) mode. Data analysis was based on MassLynx V4.2 software, (Waters Corporation, Milford, MA, USA). LC-ESI-MS/MS chromatograph and the results of metabolites determination collected by multiple reaction monitoring (MRM) mode are presented in Figure 2. Concentrations were calculated on the basis of calibration standard curves, constructed using linear regression analysis for peak area depending on concentration, and for cortisol, on the basis of factor in relation to the internal standard. Nitrogen (99.9%) from the Peak NM20ZA instrument was used as curtain gas. The sample preparation process and other parameters of the used analytical methods are described in the Table 7.

#### 4.3.3. Magnetic Resonance Techniques

Magnetic resonance spectroscopy (MRS) and imaging (MRI) were performed at MRI unit of the Department of Diagnostic Imaging at the University Hospital in Krakow, Poland with using 3 T Siemens Magnetom Vida Fit whole-body magnetic resonance scanner. MRS and MRI scans were performed on scanner with a 20-channel head coil, with strong whole-body gradients with a rise rate of 200 T/m/s on each axis and an amplitude of 45 mT/m. Magnetic resonance examinations were performed on the same day. To exclude any changes in the central nervous system, according to MRI protocol, study participants were first subjected to non-contrast brain imaging with following sequences: 3D T2 sequences in sagittal plane (spacing 0.9 mm, slice thickness 1.0 mm, Time to Echo (TE) 260 ms, Repetition Time (TR) 3200 ms, Field of View (FOV) 25 cm, Flip Angle (FA) 120°, matrix 256 × 256 pixels, scanning sequences: spin echo, space); 3D T1 sequences in sagittal plane (spacing 0.9 mm, slice thickness 1.2 mm, TE 260 ms, TR 2300 ms, FOV 25 cm, FA 9°, matrix 256 × 248 pixels, scanning sequences: gradient, mprage); 3D dark fluid sequences in sagittal plane (spacing 0.9 mm, slice thickness 1.2 mm, TE 394 ms, TR 7000 ms, Inversion Time (TI) 2050 ms, FOV 25 cm, FA 120°, matrix 256 × 288 pixels, scanning sequences: spin echo resolve); 2D DWI sequences in axial plane (spacing 3.9 mm, slice thickness 3.0 mm, TE 65 ms, TR 4120 ms, FOV 23 cm, FA 160°, b 1000 s/mm^2^, matrix 448 × 448 pixels, scanning sequences: epi).

Coronal and axial planes (spacing: 0 mm, slice thickness 1.00 mm) were reconstructed automatically from T1, T2 and dark fluid images. To planning VOI (volume of interest) position for spectroscopy and to visualize anatomical brain structure we performed three-dimensional T2 imaging. Single-voxel spectroscopy technique (SVS) was used in next step with following parameters: TR 2000 ms, TE 30 ms and 144 ms, 64 averages acquired. To record metabolites with short and long relaxation times, two echo times (TE): 30 ms and 144 ms, were used. Two symmetrically situated locations in the anterior cingulate cortex (ACC) and posterior cingulate cortex (PCC) were the source of the MRS signal. The mean VOI, adapted to the location size and positioned away from susceptibility artifacts source, was 3,4 cm^3^. LCModel [133], intended for automatic quantitation of in vivo 1H MR spectra, was used for analyzed of MRS data. Concentrations of metabolites were established on the basis of their resonance areas. Metabolites selected from the spectrum were: L-Alanine (Ala 1.48 ppm), Aspartate (Asp 3.8 ppm), Creatine (Cr 3.02 and 3.9 ppm), Glucose (Glc 3.43 and 3.8 ppm), Glutamate (Glu 2.1 and 3.7 ppm), Glutamine (Gln 2.45 and 3.7 ppm), Glutathione (GSH 3.7 ppm), Glycerophosphocholine (GPC 3.6 ppm), γ-aminobutyric acid (GABA 2.3 ppm), L-Lactate (Lac 1.33 ppm), myo-Inositol (Ins 3.6 ppm), N-Acetylaspartate (NAA 2.02 ppm), N-Acetylaspartylglutamate (NAAG 4.1 ppm), Phosphocreatine (PCr 3.02 ppm and 3.93 ppm), Phosphocholine (PCh 4.2 ppm), Taurine (Tau 3.42 ppm), scyllo-Inositol (Scyllo 3.35 ppm), macromolecule (MM09 0.9 ppm, MM12 1.2 ppm, MM14 1.4 ppm, MM17 1.7 ppm, MM20 2.0 ppm) and Lipids (Lip09 0.9 ppm, Llip13a and Lip13b 1.3 ppm, Lip20 2.0 ppm). Calculated was also the sum of concentrations of individual metabolites: Cr+PCr, Glu+Gln, NAA+NAAG, GPC+PCh, Lip13a+Lip13b, MM09+Lip09, MM20+Lip20, MM14+Lip13a+Lip13b+MM12. Additionally, we determined the concentration ratio of each metabolites to the sum of creatine and phosphocreatine total concentrations e.g. NAA/(Cr+PCr). Calculations were made for each VOI. For the reconstructed MRS spectra qualitative and quantitative analysis were performed. Examination and analysis of the MRS spectrum was performed by experienced medical physicist. Quality control, applied on a routine procedures and performed by an experienced radiologist, was carried out in relation to the signal-to-noise ratio and to the width of the spectral lines. Spectra were excluded if: there was a lack of adequate unsuppressed water acquisition, width of spectral line, compared to the overall mean for the voxel assessed for all sites and all participants, was 2 SD (standard deviations) above or signal-to-noise ratio of spectra was 2 SD below. We excluded concentrations of individual metabolites associated with Cramér Rao lower bounds (CRLB) > 20% [134].

#### 4.3.4. Statistical Methods

The significance level of the statistical tests in this analysis was set at α = 0.05. Analysis of the normality of the distributions of the numerical variables. The normality of the distributions of the variables was analysed using the Shapiro-Wilk test. Numerical variables with distributions deviating from the normal distribution were reported as median (Q1, Q3). Examination of differences within a numerical variable with a non-normal distribution between two groups was performed with the Wilcoxon rank sum test. The effect size of Wilcoxon rank sum test r was calculated as the Z statistic divided by the square root of the sample size N, r = z/√N. Interpretation values for r commonly in published literature [135] are: [0.10–0.30)—small effect), [0.30–0.5)—moderate effect, and [0.50–1.00]—large effect. Spearman’s rank correlation coefficient (rho) was used to measure the strength and direction of association between two variables. The p-values were computed using algorithm AS 89 [136]. Determination and evaluation of optimal cutpoints between control and test groups was performed using the maximization sum of sensitivity and specificity with more or equal (“≥”) direction with bootstrapping procedures. The quality of classification based on the determined cutoff point was examined using the metrics of sensitivity, specificity, accuracy, and AUC. Sensitivity, specificity, and accuracy were calculated using Formulas (1)–(3), respectively:(1)$$\mathrm{Sensitivity}=\frac{\mathrm{True}\;\mathrm{positives}}{\mathrm{True}\;\mathrm{positives}+\mathrm{False}\;\mathrm{negatives}},$$
(2)$$\mathrm{Specificity}=\frac{\mathrm{True}\;\mathrm{negatives}}{\mathrm{True}\;\mathrm{negatives}+\mathrm{False}\;\mathrm{positives}},$$
(3)$$\mathrm{Accuracy}=\frac{\mathrm{True}\;\mathrm{positives}+\mathrm{True}\;\mathrm{negatives}}{\mathrm{True}\;\mathrm{positives}+\mathrm{False}\;\mathrm{negatives}+\mathrm{True}\;\mathrm{negatives}+\mathrm{False}\;\mathrm{positives}},$$
where: true Positives (TP) are the instances that are correctly classified as positive; false negatives (FN) are the instances that are actually positive but are incorrectly classified as negative; true negatives (TN) are the instances that are correctly classified as negative; false positives (FP) are the instances that are actually negative but are incorrectly classified as positive.

In addition, the AUC metric was calculated as an integral of the ROC curve to quantify the ability of the model to discriminate between positive and negative classes at different thresholds. The evaluation of variability and out-of-sample performance was based on 1000 bootstrapping samples. A sample of the same size as the original data was drawn from the original data and replaced. Cutpoint estimation was performed for the in-bag sample and the determined cutpoint was applied to the in-bag and out-of-bag observations [137].

##### Clustering Analysis

The hierarchical cluster analysis on the principal components (HCPC) was performed to identify groups of similar observations for test scores the overall results for 14 tests. Prior to clustering, principal component analysis (PCA) was conducted to reduce the dimensionality of the data while retaining maximum variation. Hierarchical clustering was performed using Ward’s minimum variance method. The optimal number of clusters was determined using the Elbow and Silhouette methods. Differences in the results of individual tests between clusters were estimated using the V test.

##### Statistical Environment

Analyses were conducted using the R Statistical language (version 4.1.1) [138], on Windows 10 Pro 64 bit (build 19045), using the packages rio (version 0.5.29) [139], rstatix (version 0.7.1) [140], factoextra (version 1.0.7) [141], FactoMineR (version 2.6) [142], report (version 0.5.7) [143], gtsummary (version 1.6.2) [144], cutpointr (version 1.1.2) [137], ggplot2 (version 3.4.0) [145], readxl (version 1.3.1) [146], and dplyr (version 1.1.2) [147].

## 5. Conclusions

Schizophrenia as a complex disease, and the diagnostic process requires a broad look at both clinical assessment, laboratory test results, and self-reported test results. The correlations of metabolites determined in the blood and brain and test scores obtained in our results can be used in the individual process of diagnosing of schizophrenia. Changes in the concentration of metabolites in the blood such as glutamate, lactate and cortisol may indicate the onset of mental disorders, exacerbation of the disease or response to treatment. Correlations of metabolites determined in blood with brain neuroimaging parameters can also be used as a tool in searching for the cause of mental disorders. Changing concentrations of metabolites may result from disturbances in the functioning of the glutamatergic transmission or disturbances in the functioning of the hypothalamic-pituitary-adrenal axis. Clustering analysis allowed us to divide patients with schizophrenia into three groups with similar overall results for 14 self-reported tests. The distinction of three clusters of patients with schizophrenia based on the results of self-reported assessments allowed for classification of patients into groups with the most similar origin of the disease. Schizophrenia may be associated with gastrointestinal disorders, inflammatory reactions, trauma, or a disturbed response to stress. Complex disease as schizophrenia requires a comprehensive diagnostic approach selected individually for each patient. The combination of clinical assessment with laboratory test, brain neuroimaging, and the results of self-reported assessment may allow for faster diagnosis and, therefore, faster and more effective treatment of patients with schizophrenia.

## Figures and Tables

**Figure 1 ijms-25-02294-f001:**
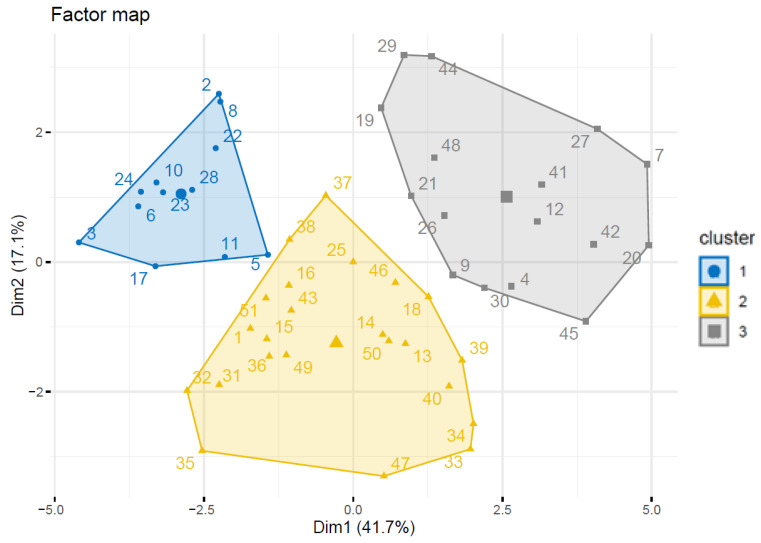
Location of clusters with test results in Dim1, Dim2 space (values within clusters indicate the ID of patients in the test group). The size of the 1, 2, and 3 clusters was n1 = 12, n2 = 23, and n3 = 16 individuals, respectively. Dim—two-dimensional space.

**Figure 2 ijms-25-02294-f002:**
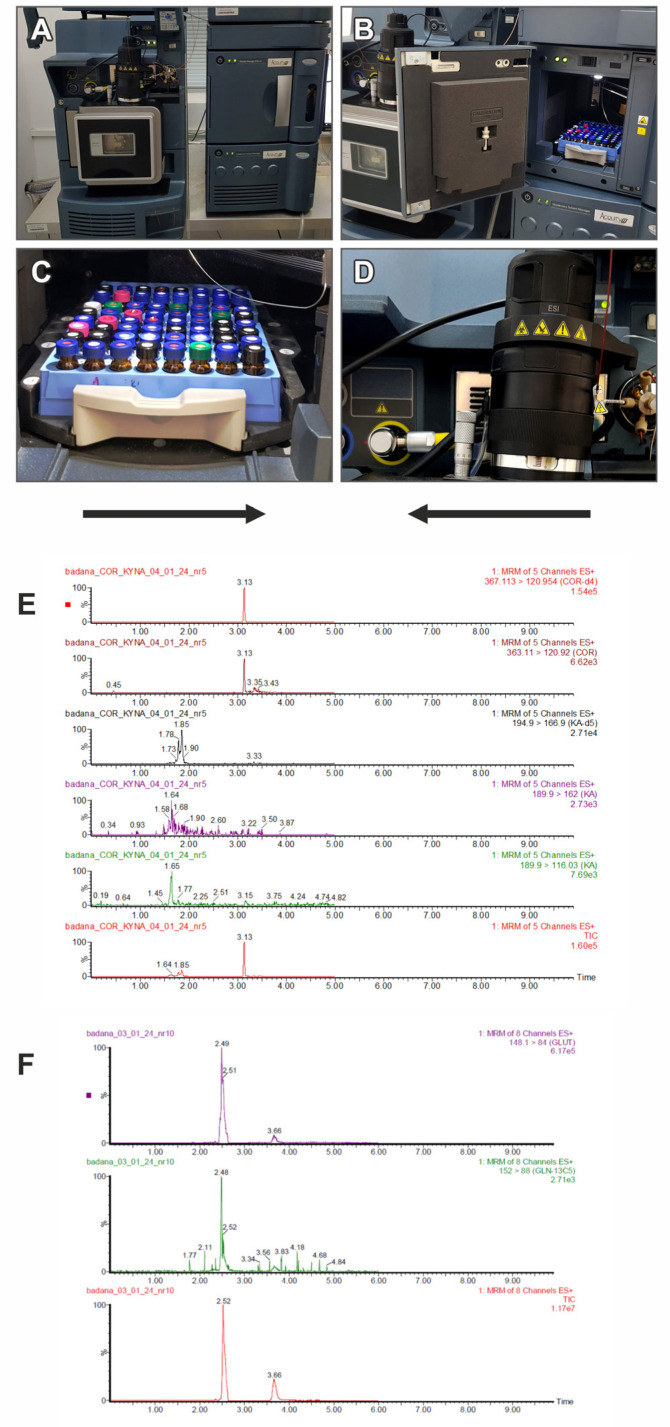
LC-ESI-MS/MS chromatograph and the results of metabolites determination collected by multiple reaction monitoring (MRM) mode. (**A**) Waters ACQUITY UPLC^®^H-Class set – liquid chromatograph with mass detector; (**B**) view of an open autosampler; (**C**) sample tray in the autosampler; (**D**) replaceable head for electrospray analysis (ESI) in an electric field; (**E**) chromatogram showing peaks for cortisol (COR) and kynureninic acid (KA) and their deuterine derivatives, the lowest chromatogram shows TIC (Total Ion Current) for the tested sample; (**F**) chromatogram for glutamine (GLN), deuterium derivative of glutamate (GLUT) and TIC for the entire sample; (**G**) TIC of the entire analyzed sample—the equivalent of a classic chromatogram.

**Table 1 ijms-25-02294-t001:** Distribution of clinical parameters for studied and control groups with estimation of differences between groups.

Characteristic	N	Group		*p ^2^*	*r*
		Control, n = 45 ^1^	Test, n = 51 ^1^		
Serotonin [µg/mL]	96	133.46 (127.63, 138.31)	130.30 (126.21, 137.87)	0.301	0.11
Alanine [µg/mL]	96	1.52 (1.23, 1.83)	1.52 (1.22, 1.76)	0.977	<0.01
Glutamic acid [µg/mL]	95	2923.96 (2139.69, 3630.11)	2367.73 (1411.55, 3246.98)	0.048	0.20
missing value		0	1		
Glutamine [µg/mL]	96	334.20 (286.22, 367.14)	354.35 (307.89, 382.01)	0.249	0.12
Cortisol [ng/mL]	96	162.34 (131.19, 184.54)	120.57 (93.37, 174.50)	0.031	0.22
Lactates [µg/mL]	94	156.98 (129.59, 177.59)	120.92 (88.37, 146.86)	<0.001	0.41
missing value		1	1		
Lactates [mmol/L]	94	1.74 (1.44, 1.97)	1.34 (0.98, 1.63)	<0.001	0.41
missing value		1	1		
Kinurenic acid [ng/mL]	95	54.80 (40.01, 65.52)	46.81 (35.55, 60.14)	0.167	0.14
missing value		0	1		
Acetic acid [nM]	96	9.56 (7.71, 11.15)	8.86 (6.19, 10.86)	0.071	0.19
Propionic acid [nM]	96	0.54 (0.42, 0.64)	0.50 (0.36, 0.60)	0.146	0.15
Butyric acid [nM]	96	0.07 (0.03, 0.10)	0.06 (0.03, 0.10)	0.675	0.04
Isobutyric acid [nM]	96	0.18 (0.14, 0.24)	0.17 (0.06, 0.24)	0.215	0.13
Valerian acid [nM]	96	0.06 (0.04, 0.13)	0.05 (0.01, 0.12)	0.145	0.15
Isovaleric acid [nM]	96	0.31 (0.23, 0.48)	0.35 (0.18, 0.52)	0.863	0.02

Note: n—sample size; *p*—the *p*-value of the statistical test; r—Pearson’s correlation coefficient; Mdn—median; Q1—first quartile (25%); Q3—third quartile (75%); µg/mL—microgram per milliliter; ng/mL nanogram per milliliter; mmol/L—millimole per liter; nM—nanomolar. ^1^ Mdn (Q1, Q3); ^2^ Wilcoxon rank sum test.

**Table 2 ijms-25-02294-t002:** Estimated optimal cutpoints with the metric performance of binary classification (studied or control group) for parameters with significant differences between groups.

Characteristic	Optimal Cutpoint	Accuracy	Sensitivity	Specificity	AUC
Cortisol [ng/mL]	131.19	0.69	0.76	0.63	0.63
Glutamic acid [µg/mL]	1671.59	0.62	0.91	0.36	0.62
Lactates [µg/mL]	112.15	0.70	0.95	0.48	0.74

Note: AUC—area under the receiver operating characteristic curve; µg/mL—microgram per milliliter; ng/mL nanogram per milliliter.

**Table 3 ijms-25-02294-t003:** The results of the correlation analysis of lactates, glutamic acid and cortisol concentrations with the results of clinical evaluation and self-reported questionnaires.

Questionnaire	Lactates [µg/mL]			Glutamic Acid [µg/mL]			Cortisol [ng/mL]		
	n_pairs_	rho	*p*	n_pairs_	rho	*p*	n_pairs_	rho	*p*
Positive symptoms	49	0.09	0.507	49	0.24	0.093	50	0.14	0.322
Negative symptoms	49	-0.09	0.547	49	0.13	0.360	50	−0.04	0.773
Disorganized speech	49	0.10	0.500	49	0.18	0.205	50	0.17	0.242
Uncontrolled hostility excitement	49	0.12	0.412	49	0.34	0.016	50	0.07	0.628
Anxiety depression	49	0.04	0.806	49	0.15	0.290	50	0.06	0.658
P1–P7	49	0.09	0.536	49	0.23	0.100	50	0.07	0.614
N1–N7	49	−0.06	0.701	49	0.09	0.511	50	−0.01	0.925
G1–G16	49	0.09	0.546	49	0.30	0.030	50	0.17	0.238
Total score	49	0.04	0.790	49	0.25	0.075	50	0.13	0.353
Total score	49	−0.06	0.699	49	0.18	0.198	50	0.01	0.960
Total score	49	0.23	0.108	49	0.26	0.062	50	−0.09	0.548
Total score	49	0.14	0.340	49	0.22	0.128	50	−0.06	0.674
Total score	49	−0.15	0.288	49	−0.23	0.106	50	0.03	0.811
Total score	46	0.21	0.145	46	0.17	0.224	47	0.07	0.647
Total score	45	0.14	0.313	45	0.17	0.229	46	0.08	0.581
Total score	45	0.29	0.038	45	0.26	0.068	46	−0.02	0.900
SSZ	38	0.07	0.609	38	0.18	0.216	39	0.09	0.513
SSE	38	0.08	0.596	38	0.17	0.234	39	0.09	0.532
SSU	38	−0.15	0.294	38	−0.06	0.689	39	−0.07	0.649
ACZ	38	−0.16	0.251	38	0.01	0.919	39	−0.13	0.349
PKT	38	−0.03	0.852	38	−0.19	0.175	39	−0.02	0.909
Total score	38	0.04	0.789	38	0.17	0.225	39	0.03	0.852
Total score	45	0.30	0.032	45	0.33	0.019	46	0.09	0.552
Reexperiencing trauma	41	0.09	0.547	41	0.34	0.014	42	0.16	0.263
Avoidance	41	0.39	0.005	41	0.36	0.009	42	0.13	0.347
Threat	41	0.18	0.206	41	0.25	0.071	42	0.17	0.231
Affective dysregulation	41	−0.23	0.099	41	0.00	0.974	42	−0.01	0.921
Negative self-concept	41	−0.03	0.853	41	0.06	0.700	42	0.06	0.683
Disturbance relationships	41	0.04	0.779	41	0.05	0.711	42	0.03	0.847
PTSDFI	41	0.07	0.606	41	−0.03	0.814	42	0.08	0.579
DSOFI	41	−0.01	0.929	41	−0.05	0.722	42	0.06	0.681
PTSD	41	0.26	0.069	41	0.36	0.010	42	0.16	0.264
DSO	41	−0.05	0.735	41	0.07	0.601	42	−0.02	0.904
Total score	41	0.10	0.470	41	0.10	0.475	42	0.05	0.739
Emotional abuse	41	0.13	0.376	41	0.02	0.865	42	0.01	0.924
Physical abuse	41	−0.20	0.168	41	−0.18	0.200	42	−0.07	0.650
sexual abuse	41	0.12	0.411	41	0.03	0.832	42	−0.22	0.129
Emotional neglect	41	−0.12	0.393	41	−0.09	0.525	42	−0.15	0.285
Physical neglect	41	0.01	0.949	41	0.16	0.270	42	−0.28	0.045
Denial	41	0.01	0.938	41	0.00	0.998	42	0.08	0.593
Total score	41	−0.08	0.585	41	−0.05	0.731	42	−0.08	0.567
Total score	40	0.03	0.825	40	0.01	0.940	41	−0.06	0.680
Total score	40	0.02	0.866	40	−0.08	0.575	41	−0.24	0.088

Note: n—sample size; p—the *p*-value of the statistical test; rho—Spearman’s rank correlation coefficient; PANSS—Positive and Negative Syndrome Scale; P1–P7—Positive Scale; N1–N7—Negative Scale; G1–G16—General Psychopathology Scale; SSZ—the task-oriented style; SSE—the emotional style; SSU—avoidance-focused style; ACZ—involvement in displacement activit; PKT—search for social contact; PTSDFI—post traumatic stress disorder functional impairment; DSOFI—disturbances in self-organization functional impairment; PTSD—post traumatic stress disorder; DSO—disturbances in self-organization.

**Table 4 ijms-25-02294-t004:** The results of correlation analysis between lactates, glutamic acid, cortisol and selected laboratory, clinical and brain parameters.

Characteristic	Lactates [µg/mL]			Glutamic acid [µg/mL]			Cortisol [ng/mL]		
	n_pairs_	rho	*p*	n_pairs_	rho	*p*	n_pairs_	rho	*p*
WBC [×10^3^/µL]	94	−0.12	0.226	95	−0.07	0.522	96	−0.18	0.074
NEUT [×10^3^/µL]	94	−0.21	0.044	95	−0.12	0.248	96	−0.16	0.120
Re-Lymph [×10^3^/µL]	94	−0.35	<0.001	95	−0.07	0.498	96	−0.24	0.020
IG [×10^3^/µL]	94	−0.11	0.284	95	0.07	0.496	96	−0.22	0.035
NEUT [%]	94	−0.29	0.005	95	−0.21	0.041	96	−0.12	0.225
Lymph [%]	94	0.26	0.012	95	0.15	0.133	96	0.06	0.549
Re-Lymph [%]	94	−0.33	0.001	95	−0.04	0.734	96	−0.18	0.073
EO [%]	94	0.16	0.115	95	0.32	0.001	96	−0.01	0.951
BASO [%]	94	0.12	0.244	95	0.11	0.273	96	0.10	0.328
RBC [×10^6^/µL]	94	−0.06	0.585	95	0.01	0.894	96	−0.14	0.168
Hgb [g/dl]	94	−0.08	0.448	95	0.01	0.944	96	−0.12	0.253
Hct [%]	94	−0.09	0.406	95	−0.05	0.596	96	−0.17	0.092
Macrocytes [%]	94	−0.03	0.777	95	0.03	0.768	96	−0.08	0.435
K^+^ [mmol/L]	94	−0.09	0.378	95	0.00	0.983	96	−0.13	0.200
Glucose [mmol/L]	94	−0.33	0.001	95	−0.40	<0.001	96	−0.01	0.924
Uric acid [µmol/L]	94	−0.11	0.266	95	0.03	0.809	96	−0.11	0.271
Cholesterol HDL [µmol/L]	93	0.26	0.011	94	0.01	0.903	95	0.21	0.037
Triglycerides [µmol/L]	93	−0.03	0.796	94	0.04	0.677	95	−0.10	0.337
FT4 [pmol/L]	94	0.27	0.008	95	0.11	0.295	96	0.00	0.992
DHEA-S [µmol/L]	94	−0.16	0.131	95	−0.04	0.681	96	0.09	0.408
Insulin [µU/mL]	94	0.00	0.993	95	0.04	0.709	96	−0.10	0.342
HOMA-IR	92	−0.04	0.674	93	−0.02	0.853	94	−0.10	0.309
Creatine conc [×10^6^]	91	0.24	0.019	92	0.13	0.217	93	0.19	0.062
Glucose Cr+PCr	91	−0.10	0.329	92	0.02	0.828	93	−0.09	0.404
Glucose Cr+PCr	91	−0.15	0.144	92	0.05	0.630	93	−0.14	0.174
Glutamine Cr+PCr	91	0.17	0.098	92	0.15	0.152	93	0.28	0.005
Glutamine Cr+PCr	91	0.09	0.364	92	0.20	0.049	93	0.17	0.097
Inositol conc [×10^6^]	91	0.06	0.586	92	0.01	0.928	93	0.04	0.722
N-Acetylaspartate conc [×10^6^]	91	0.15	0.148	92	0.04	0.729	93	0.23	0.026
Cr+PCr conc [×10^6^]	91	0.27	0.008	92	0.05	0.604	93	0.30	0.003
Glu+Gln conc [×10^6^]	91	0.33	0.001	92	0.08	0.465	93	0.21	0.042
Glu+Gln Cr+PCr	91	0.19	0.059	92	0.10	0.334	93	0.02	0.869
Phosphocreatine conc	91	0.19	0.067	92	0.08	0.467	93	0.10	0.344
Glutamate conc	91	0.26	0.009	92	−0.04	0.707	93	0.13	0.193
Glu/Cr+PCr	91	0.20	0.048	92	−0.09	0.403	93	0.04	0.683
N-Acetylaspartate conc.	91	0.30	0.003	92	0.03	0.787	93	0.07	0.496
NAA/Cr+PCr	91	0.21	0.039	92	0.01	0.900	93	−0.07	0.504
N-Acetyaspartate + N-Acetylspartylglutamate conc	91	0.27	0.009	92	−0.03	0.785	93	0.15	0.150
NAA+NAAG/Cr+PCr	91	0.19	0.069	92	0.00	0.965	93	0.04	0.713
Cr+PCr conc	91	0.23	0.026	92	0.04	0.728	93	0.14	0.164
Glu+Gln conc	91	0.23	0.025	92	0.00	0.965	93	0.12	0.249
Glu+GlnCr+PCr	91	0.13	0.197	92	0.00	0.999	93	0.06	0.537
Phosphocreatine conc [×10^3^]	91	−0.01	0.960	92	−0.05	0.618	93	0.16	0.111
N-Acetylaspartate conc [×10^6^]	91	0.04	0.727	92	0.02	0.867	93	0.09	0.371
Taurine conc [×10^6^]	91	−0.18	0.087	92	−0.02	0.873	93	−0.22	0.028
Taurine Cr+PCr [×10^3^]	91	−0.18	0.081	92	−0.01	0.896	93	−0.24	0.020
Cr+PCr conc [×10^6^]	91	0.12	0.242	92	0.02	0.833	93	0.20	0.056
Lip20 conc [×10^6^]	91	0.13	0.222	92	0.13	0.205	93	0.08	0.427
Lip20 Cr+PCr [×10^3^]	91	0.11	0.265	92	0.13	0.223	93	0.07	0.495
L-alanine conc	91	0.07	0.528	92	−0.10	0.350	93	0.21	**0.045**
Creatine Cr conc.	91	−0.06	0.539	92	−0.07	0.475	93	−0.14	0.189
Creatine, Cr/(Cr+PCr)	91	−0.05	0.599	92	−0.10	0.354	93	−0.12	0.250
Phosphocreatine, PCr	91	0.03	0.769	92	0.04	0.694	93	0.11	0.283
Phosphocreatine, PCr/(Cr+PCr)	91	0.05	0.599	92	0.010	0.354	93	0.12	0.250
Scylloinositol conc.	91	−0.04	0.692	92	0.08	0.451	93	−0.10	0.321
Scylloinositol/(Cr+PCr)	91	−0.04	0.703	92	0.09	0.387	93	−0.10	0.322
GPC+PCh/Cr+PCr	91	−0.12	0.244	92	0.02	0.848	93	0.02	0.860

Note: n—sample size; *p*—the *p*-value of the statistical test; rho—Spearman’s rank correlation coefficient; WBC—white blood cells; NEUT—neutrophils; Re-Lymph—reactive lymphocytes; IG—immature granulocytes; Lymph—lymphocytes; EO—eosinophils; BASO—basophils; RBC—red blood cells; Hgb—hemoglobin; Hct—hematocrit; K^+^—potassium; FT4—free thyroxine; DHEA-S—sulfated form of dehydroepiandrosterone; HOMA-IR—homeostasis model assessment of insulin resistance; Cr—creatine; PCr—phosphocreatine; Glu—glutamate; Gln—glutamine; NAA—N-acetylaspartate; NAAG—N-acetylspartylglutamate; GPC—glycerophosphocholine; PCh—phosphocholine.

**Table 5 ijms-25-02294-t005:** Characteristics of the profiles of individuals according to cluster membership.

Test	V test	M_Cluster_	M_test group_	SD_Cluster_	SD_test group_	*p*
Cluster 1 (n = 12)						
GAF	4.25	68.92	52.34	10.48	15.31	<0.001
Gastrointestinal symptoms	−2.36	9.50	18.83	4.68	15.49	0.018
ITQ	−2.44	23.75	33.83	11.35	16.19	0.015
CTQ	−2.46	57.08	66.45	8.23	14.95	0.014
GHQ-28	−2.51	22.42	31.85	7.94	14.77	0.012
STAI	−2.57	81.83	95.83	11.98	21.34	0.010
CALGARY	−3.14	3.67	8.58	3.25	6.14	0.002
BDI II	−3.34	7.33	18.76	5.88	13.41	0.001
SAPS	−4.69	8.50	35.28	7.31	22.38	<0.001
SANS	−5.06	20.25	51.28	10.10	24.05	<0.001
PANSS	−5.08	47.83	78.20	19.58	23.46	<0.001
Cluster 2 (n = 23)						
SANS	2.88	62.10	51.28	13.78	24.05	0.004
PANSS	2.60	87.70	78.20	13.39	23.46	0.009
SAPS	2.50	44.01	35.28	22.25	22.38	0.012
ECR-RS	−2.09	47.44	52.76	12.53	16.34	0.037
CTQ	−2.21	61.29	66.45	9.68	14.95	0.027
STAI	−2.34	88.05	95.83	17.32	21.34	0.020
BDI II	−2.36	13.82	18.76	7.88	13.41	0.018
ITQ	−3.02	26.20	33.83	9.90	16.19	0.003
GAF	−3.03	45.10	52.34	11.55	15.31	0.002
Cluster 3 (n = 16)						
BDI II	5.59	34.44	18.76	8.90	13.41	<0.001
ITQ	5.47	52.36	33.83	9.75	16.19	<0.001
STAI	4.86	117.50	95.83	14.45	21.34	<0.001
CTQ	4.62	80.90	66.45	14.62	14.95	<0.001
GHQ 28	4.34	45.25	31.85	13.13	14.77	<0.001
ECR-RS	3.94	66.22	52.76	15.23	16.34	<0.001
Gastrointestinal symptoms	3.55	30.31	18.83	20.58	15.49	<0.001
TEC PL	2.34	28.08	21.32	16.13	13.80	0.019
CALGARY	2.23	11.44	8.58	6.22	6.14	0.026
CISS	2.00	149.05	141.87	19.63	17.15	0.046

Note: V test—the V-test statistic; M_Cluster_—the mean for cluster; M_test group_—the mean for the entire test group; SD_Cluster_—the SD for cluster; SD_test group_ the SD for the entire test group; *p*—the *p*-value of the statistical test; PANSS—Positive and Negative Syndrome Scale; SANS—Scale for Assessment of Negative Symptoms; SAPS—Scale for Assessment of Positive Symptoms; CALGARY—Calgary Depression Scale for Schizophrenia; GAF—Global Assessment of Functioning; BDI II—the Beck Depression Inventory; STAI—State and Trait Anxiety Inventory, CISS—Coping Inventory for Stressful Situations; GHQ 28—General Health Questionnaire-28, ITQ—International Trauma Questionnaire; CTQ—Childhood Trauma Questionnaire; ECR-RS—Experiences in Close Relationships-Revised Short; TEC PL—Traumatic Experiences Checklist Poland.

**Table 6 ijms-25-02294-t006:** Distributions of clinical parameters for the entire sample.

Characteristic	N	Distribition ^1^
Serotonin [µg/mL]	96	131.32 (126.66, 138.05)
Alanine [µg/mL]	96	1.52 (1.23, 1.83)
Glutamic acid [µg/mL]	95	2732.44 (1785.60, 3575.55)
Glutamic acid [µg/mL]	96	341.84 (289.56, 380.60)
Cortisol [ng/mL]	96	138.96 (106.69, 184.22)
Lactates [µg/mL]	94	135.33 (111.05, 167.62)
Lactates [mmol/L]	94	1.50 (1.23, 1.86)
Kinurenic acid [ng/mL]	95	48.76 (37.05, 64.44)
Acetic acid [nM]	96	8.96 (6.78, 11.11)
Propionic acid [nM]	96	0.53 (0.38, 0.62)
Butyric acid [nM]	96	0.07 (0.03, 0.10)
Isobutyric acid [nM]	96	0.18 (0.11, 0.24)
Valerian acid [nM]	96	0.06 (0.02, 0.13)
Isovaleric acid [nM]	96	0.34 (0.21, 0.49)

Note: N—sample size; Mdn—median; Q1—first quartile (25%); Q3—third quartile (75%); µg/mL—microgram per milliliter; ng/mL nanogram per milliliter; mmol/L—millimole per liter; nM—nanomolar. ^1^ Mdn (Q1, Q3).

**Table 7 ijms-25-02294-t007:** Description of LCMS analyzes used to determine analytical parameters in peripheral blood.

	LC-ESI-MS/MS	LCMS-SCFA	LC-ESI-MS/MS Cortisol
Sample preparation	4 µL of internal standard (methanolic solution of the tested compounds with a concentration of 500 µg/mL) was added to 100 µL of sample (serum). Protein was precipitated by adding 100 μL of ice-cold acetonitrile. After mixing, the samples were centrifuged (8000 rpm, 5 min, 15 °C). The supernatant was collected and placed in chromatographic vessels.	60 µL of a mixture of methanol and water (1:1, *v*/*v*) was added to 20 µL of serum, the samples were vortexed, then centrifuged (14000 rpm, 10 min, 4 °C), 40 µL of the supernatant was collected. Derivatization: 40 µL of water was added to 40 µL of the supernatant, then 10 µL of 0.1 M BHA and 10 µL of 0.25 M EDC. After mixing, the samples were incubated for 1 h at 25 °C with constant stirring. The derivatization process was completed by adding 100 µL of 50% methanol and 600 µL of dichloromethane. The samples were then mixed and centrifuged (8000 rpm, 10 min). The organic phase was collected and evaporated to dryness under nitrogen and then dissolved in 30 µL of 50% methanol.	4 µL of internal standard (methanolic solution containing cortisol with a concentration of 500 µg/mL) was added to 100 µL of sample (serum). Protein was precipitated by adding 100 μL of ice-cold acetonitrile. After mixing, the samples were centrifuged (8000 rpm, 5 min, 15 °C). The supernatant was collected and placed in chromatographic vessels.
Chromatography column	ZIC^®^-HILIC (5 µm, 200 Å, 150 × 21.2 mm; Merck, Darmstadt, Germany), thermostated at 40 °C	Kinetex (1.7 µm Biphenyl, 100 Å; 100 × 2.1 mm; Phenomenex Companies Worldwide, Torrance, CA, USA)	Kinetex (2.6 µm Biphenyl 100 Å, 100 × 2.1 mm; Phenomenex Companies Worldwide, Torrance, CA, USA), thermostated at 40 °C
Mobile phases	Phase A: 0.1% aqueous HCOOH solution; Phase B: 0.1% HCOOH in acetonitrile	Phase A: 0.1% HCOOH + 10 mM NH4COOH; Phase B: 0.1% HCOOH in a mixture of methanol: isopropanol (9: 1 *v*/*v*)	Phase A: 0.1% aqueous HCOOH solution; Phase B: methanol
Phase flow rate	0.6 mL/min	0.4 mL/min	0.4 mL/min
Separation program	Gradient separation program: 0–0.2 min, isocratic gradient 5.0% of phase A; 0.2–1.5 min, linear gradient 5–55% of phase A; 1.5–3.1 min, isocratic gradient 55% of phase A; 3.1–4.5 min, linear gradient 55.0–5.0% of phase A; 4.5–6 min, isocratic gradient 5.0% of phase A	Gradient separation program: 0–4 min, 68.0–40.0% of phase A, 4–4.8 min, 35% of phase A, 4.8–4.9 min, 2% of phase A; after this time, isocratic conditions were achieved (4.9–5.2 min), and then the initial conditions were returned—68% of phase A (5.2–5.3 min)	Gradient separation program: 0–0.4 min, isocratic gradient 5.0% of phase B; 0.4–0.5 min, linear gradient 5–50% of phase B; 0.5–1.9 min, isocratic gradient 50% of phase B; 1.9–2.0 min, linear gradient 100.0% of phase B; 2.0–2.9 min, isocratic gradient 100.0% of phase B; 2.9–4.0 min, linear gradient 100–5.0% of phase B; 4.0–5.0 min, isocratic gradient 5.0% of phase B
Injection volume	4 µL	4 µL	2 µL
Time of single sample analysis	6 min	5.3 min	5 min
Retention time	ALA, ALA-d4—2.44 min; SER, SER-d4—2.35 min; GLUT, GLUT-C13—2.40 min; LA—0.99 min	acetic acid—1.4 min; propionic acid—1.9 min; isobutyric acid—2.59 min; butyric acid—2.8 min; isovaleric acid—3.2 min; valeric acid—3.5 min	COR, COR-d4—3.39 min
Ion source parameters	IS: 5500 V; atomizing gas (gas 1): 30 psi; turbo gas (gas 2): 20 psi; TEM: 550 °C; CUR: 30 psi	IS: 5400 V; atomizing gas (gas 1): 30 psi; turbo gas (gas 2): 20 psi; TEM: 550 °C; CUR: 30 psi	IS: 5500 V; atomizing gas (gas 1): 30 psi; turbo gas (gas 2): 20 psi; TEM) 300 °C; CUR: 30 psi
Ion pairs	ALA: *m*/*z* = 90.09/44.9; ALA-d4: *m*/*z* = 93.93/47.97; GLUT: *m*/*z* = 148.0/84.0; GLUT-C13: *m*/*z* = 153.0/89.0; SER: *m*/*z* = 177.0/119.0; SER-d4: *m*/*z* = 182.83/119.76; LA: *m*/*z* = 88.84/43.948	acetic acid: *m*/*z* = 166.1/91.1; propionic acid: *m*/*z* = 180.1/91.1; butyric and isobutyric acid: 194.1/91.1; valeric and isovaleric acid: *m*/*z* = 208.1/124.1	COR: *m*/*z* = 363.11/120.92; COR-d4: *m*/*z* = 367.11/120.95

Note: LC-ESI-MS/MS—Liquid Chromatography-Electrospray Ionization-Mass Spectrometry; SCFA—Short Chain Fatty Acids; ALA-d4—alanine; SER, SER-d4—serotonin; GLUT, GLUT-C13—glutamic acid; LA—lactates; COR, COR-d4—cortisol; m/z—mass to charge ratio; BHA—butylated hydroxyanisole; EDC—1-Ethyl-3-3-dimethylaminopropyl]carbodiimide hydrochloride; rpm—revolutions per minute; *v*/*v*—volume per volume; min—minute; Å—Angstrem; µL—microliter; µm—micrometer; mM—millimole; psi—pound per square inch; µg/mL—microgram per milliliter; mL/min—milliliter per minute; HCOOH—formic acid; NH4COOH—ammonium formate; CA—California; USA—United States of America; IS—voltage of spray ionization; TEM—nebulizer temperature; CUR—curtain gas.

## Data Availability

The data presented in this study are available on request from the corresponding author.

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
