# Peer review of "Association of Blood Metabolomics Biomarkers with Brain Metabolites and Patient-Reported Outcomes as a New Approach in Individualized Diagnosis of Schizophrenia"

_ijms, 2024, doi:10.3390/ijms25042294_

Round 1

Reviewer 1 Report

Comments and Suggestions for Authors

In this paper, Krzyściak et al. investigated serum metabolites using LC-ESI-MS/MS in patients with schizophrenia, identifying that cortisol, glutamate, and lactate correlate with laboratory test results, imaging results, and clinical assessment results. The topic of this study is meaningful and the manuscript is well-written, but there are some issues with it. Here are some comments on this study:

1.        In the manuscript the line numbers are missing, it would be hard to point out the questions.

2.        Figure 1 needs to have labels A,B, and C… and details should also be included in the legend.

3.        Table 1, what is the meaning of missing value?

4.        The paragraph structure of the manuscript needs to be restructured to avoid 1 or 2 sentences in a separate paragraph, which could be combined.

5.        Is it possible to use the three metabolites cortisol, glutamate and lactate for modeling the diagnosis of schizophrenia? It may improve the Accuracy and AUC.

6.        References need to be accurately cited, such as “Van den Heuvel et al. analyzed the level of hair cortisol concentration by liquid chromatography tandem mass spectrometry in a group of patients with schizophrenia, obtaining a significantly lower concentration of this parameter compared to the control group” needs a citation.

7.        ng/mL or ng/ml needs to be consistent.

Author Response

                                                                                                                                                                                                January 19th, 2024                  

Letter to the Reviewer

Dear Reviewer,

please find attached a corrected version of our paper " Association of Blood Metabolomics Biomarkers with Brain Metabolites and Patient-Reported Outcomes as a New Approach in Individualized Diagnosis of Schizophrenia.“, an original paper by Krzyściak et al. We would like to thank the Reviewer for their generally positive assessment of our manuscript and for their helpful comments. We have tried to follow all of them and we have been able to fulfill almost all the Reviewers’ requirements. We believe this has allowed us to improve the quality of the paper.

A detailed list of changes introduced to the manuscript in response to specific issues raised by the Reviewers is given below.

Actions taken are marked in the manuscript in “track changes” mode.

Reviewers' Comments to the Authors:

Reviewer #1

In this paper, Krzyściak et al. investigated serum metabolites using LC-ESI-MS/MS in patients with schizophrenia, identifying that cortisol, glutamate, and lactate correlate with laboratory test results, imaging results, and clinical assessment results. The topic of this study is meaningful and the manuscript is well-written, but there are some issues with it. Here are some comments on this study:

  1. In the manuscript the line numbers are missing, it would be hard to point out the questions.

Thank You for drawing attention to this issue, we have introduced line numbering in the manuscript.

  1. Figure 1 needs to have labels A,B, and C… and details should also be included in the legend.

We have improved Figure 1 by adding appropriate tags and explaining them in the legend.

  1. Table 1, what is the meaning of missing value?

These are the missing numerical values.

  1. The paragraph structure of the manuscript needs to be restructured to avoid 1 or 2 sentences in a separate paragraph, which could be combined.

We combined short paragraphs into longer text fragments. We thank the Reviewer for this suggestion.

  1. Is it possible to use the three metabolites cortisol, glutamate and lactate for modeling the diagnosis of schizophrenia? It may improve the Accuracy and AUC.

This is beyond the scope of this manuscript aim. The manuscript concerns on association of blood metabolomics biomarkers with brain metabolites and patient-reported outcomes as a new approach in individualized diagnosis of schizophrenia

  1. References need to be accurately cited, such as “Van den Heuvel et al. analyzed the level of hair cortisol concentration by liquid chromatography tandem mass spectrometry in a group of patients with schizophrenia, obtaining a significantly lower concentration of this parameter compared to the control group” needs a citation.

The indicated literature was cited in the text after the discussion of the results obtained by the authors of the indicated work. Following the Reviewer's suggestion, we have also cited the given work in the place indicated by the Reviewer.

  1. ng/mL or ng/ml needs to be consistent.

Thank You for your comment, we have improved the units in the text to make them consistent.

Reviewer 2 Report

Comments and Suggestions for Authors

GENERAL COMENTARIES

The article is too long. The manuscript is very dense, and hard to read through.

I would recommend the Authors to divide it in two or three different, smaller articles.

The lines of the article should be numbered for an easier reading and review.

Some acronyms are never explained. Others are explained more than once. Please explain all acronyms in a standardized way: whenever the Authors use them for the first time. And after that never use the words, but only the acronyms.

TITLE

I would not use the word “new”, and much less the word “schizophrenia”.

Delete “new” as I believe there are already other studies looking for what the Authors have been looking for. Instead of “new” I would use “possible new”. Instead of schizophrenia I would use “schizophrenia like psychosis”.

AFFILIATIONS

I would not include English translations or instates. I would prefer to see those in original languages: Polish and Italian.

I would not include here any snail mail address.

I would not include all electronic addresses here. Please provide just one electronic address for one corresponding author.

ABSTRACT

Never, but never, use any of these words to characterize schizophrenia: disease, illness, sickness.

Please leave no acronym unexplained, eg describe what means “LC-ESI-MS/MS”.

KEYWORDS

Please use no acronyms at this section, eg “MR”.

Add “psychosis” to this list.

INTRODUCTION

Again, schizophrenia is not a “mental illness”. Schizophrenia is a syndrome, the greatest imitated syndrome in psychiatry, mimicked by many different conditions in medicine. Schizophrenia is not even a mental problem. It is indeed a psychiatric disorder, a syndrome that can be caused by many systemic, infectious, autoimmune, neurologic (etc.) conditions, eg “secondary schizophrenia”, “schizophrenia-like psychosis” or “pseudo-schizophrenia”. Only, when clinicians or researchers find no cause for psychosis they can use the concept of schizophrenia, primary schizophrenia, true schizophrenia or idiopathic schizophrenia. Please, read more in this subject here:

https://pubmed.ncbi.nlm.nih.gov/33099372/

https://pubmed.ncbi.nlm.nih.gov/34978980/

https://pubmed.ncbi.nlm.nih.gov/35523150/

https://pubmed.ncbi.nlm.nih.gov/32220147/

Whenever using “United States” the Authors should specify if they are writing about “United States of America” or other country “United States of Mexico”?

Once again, do not describe schizophrenia as “disease”.

The acronyms “IMS-MS”, “LC-ESI-MS/MS”, “WHO”, “MRI”, “MRS”, are not clearly explained. I mean, I find no correlation with words and letters.

It is not only the “polygenic nature of schizophrenia” that troubles excellent studies like the Authors presents here. It is also the wrong concept that “schizophrenia is a genetic disease”. It is not. Schizophrenia is, indeed pseudogenetic syndrome. Please read more here:

https://pubmed.ncbi.nlm.nih.gov/31200193/

MATERIALS AND METHODS

I cannot understand why RESULTS and DISCUSSION came before METHODS in the article structure.

All patients should have been submitted to 24 hour electroencephalography to exclude temporal lobe epilepsy, plus lumbar puncture to exclude encephalitis. The Authors should also have done this approach to all control subjects.

The Authors use too many self-report instruments, which fail a lot whenever used by patients with psychosis.

The Authors should never include patients with less than 18 years old. I would also never include patients with more than 65 years old. Adult age goes from18 to 65. Authors should be strict with this definition. Psychotic patients with 13 years old can later become schizoaffective, if a manic episode shows up later. It is very risky to contaminate a adult sample with teenagers or elderly. Another example: elderly cold contaminate sample with dementia. Schizophrenia is a syndrome of adulthood so it should be studied in adults. It is a completely different thing to make a study of psychosis among the teenagers or the elderly, as there are many other bias.

People using caffeine and/or tobacco should be studied in different groups, as these drugs have a lot of effects on symptoms and treatment.

Please provide chlorpromazine equivalent doses for all patients. It is always an interesting variable in studies involving psychotic and medicated patients.

Please, explain all acronyms, such as “K+”, “Na+”, “Mg2+”, “HOMA-IR”, “XN-9100”, “Corp.”, “PRO”, “PRO/e801”, “AG”, “BN II”, “Henkestr.”, “ACQUITY UPLC”, “MA”, “USA”, “NM20ZA”, “3T”, “T/m/s”, “3D”, “2D DWI”, “s/mm2” etc.

Please do not provide address for companies such as “Henkestr. 127, 91052 Erlangen, Germany”

I believe instead of “K+, Na+, Mg2+” it should be “K+, Na+, Mg2+”.

Please use Trade Mark (™) and Registered (®) symbols whenever adequate.

RESULTS

I cannot understand why RESULTS come before METHODS in the article structure.

I believe “ug/ml” is a misprint. Please use the Greek letter “µ” (mu), if you mean microgram.

TABLES

Please explain following acronyms: “nM”, “ng/ml”, “mmol/l”, “nM”, “Mdn”, “Q1”, “Q3”, etc.

I believe “ug/ml” is a misprint. Please use the Greek letter “µ” (mu), if you mean microgram.

Table 3, Table 4  and Table 7 are way too long. Please make it shorter. Other options would be to produce a horizontal table instead of a vertical one. Or divide it in two smaller tables.

I would recommend moving Table 7 to the RESULTS section.

Table 8 has too much text. Please reduce it to the necessary minimum.  And explain all acronyms at the bottom.

FIGURES

I believe figures are out of place.

Images of laboratory machinery should be part of METHODS section, not INTRODUCTION.

Figure 1 at Page 4 is at very low quality, and I think they belong to the METHODS or RESULTS section, not INTRODUCTION.

Please explain the meaning of colors in Figure 2.

DISCUSSION

I am very disappointed to read this fragment “antipsychotic drugs (fluoxetine or S-citalopram).” Fluoxetine is an antidepressant drug, with no antipsychotic effect. I do not know what the Authors mean with “S-citalopram”. I know and use escitalopram and citalopram in my clinical practice for 15 years, but I am not aware of a drug called “S-citalopram”.

Again, where I read “level of glutamate decreased after 10 days of therapy with both anti-psychotic drugs.” It should be antidepressant instead of “anti-psychotic”.

Olanzapine is not a neuroleptic. It is instead a second generation antipsychotic with virtually no “extrapyramidal symptoms”. Haloperidol is a first generation antipsychotic, which is synonymous with neuroleptic. Beware: only first generation antipsychotics are neuroleptics. Second generation antipsychotics (eg olanzapine) and third generation antipsychotic (eg aripiprazole) are not neuroleptics.

Limitations should be discussed here in a serious way. The authors should acknowledge that they did not make a proper exclusion of organic cause for psychosis in patients, nor brain anomaly in controls.

Explain all acronyms such as “1H”, “ug/ml”, “G1-G16”, “TSH”, “ITQ”, “ICD-11”, “pH”, “H+”, “i.e.”, “BBB”, “mmol/l”.

I believe it should be “H+” instead of “H+”, and “O2” instead of “O2”.

CONCLUSIONS

Authors should be more careful in this section. Information taken from “self-report” questionnaires can be very tricky to interpret in psychotic patients, even more in patients with schizophrenia. Anosognosia or lack of insight can be an important bias, whenever interpreting this kind of data.

REFERENCES

This section is way too long.

Again I would recommend Authors to divide the article in two or even three smaller articles.

Author Response

January 19th, 2024                 

Letter to the Reviewer

Dear Reviewer,

Please find attached a corrected version of our paper " Association of Blood Metabolomics Biomarkers with Brain Metabolites and Patient-Reported Outcomes as a New Approach in Individualized Diagnosis of Schizophrenia.“, an original paper by Krzyściak et al. We would like to thank the Reviewer for their generally positive assessment of our manuscript and their helpful comments. We have tried to follow all of them and we have been able to fulfill almost all the Reviewers’ requirements. We believe this has allowed us to improve the quality of the paper.

A detailed list of changes introduced to the manuscript in response to specific issues raised by the Reviewers is given below.

Actions taken are marked in the manuscript in “track changes” mode.

Reviewers' Comments to the Authors:

Reviewer #2

GENERAL COMENTARIES

The article is too long. The manuscript is very dense, and hard to read through.

I would recommend the Authors to divide it in two or three different, smaller articles.

We agree with the author of the review that the text is long. However, the results we obtained would be difficult to comprehensively analyze in separate texts. The text presents a certain approach to clinical data, only a comprehensive analysis of which can allow for the assessment of the correctness of the conclusions drawn.

The lines of the article should be numbered for easier reading and review.

Thank You for drawing attention to this issue, we have introduced line numbering in the manuscript.

Some acronyms are never explained. Others are explained more than once. Please explain all acronyms in a standardized way: whenever the Authors use them for the first time. And after that never use the words, but only the acronyms.

Thank You  for your comment. We have explained all the abbreviations used in the text.

TITLE

I would not use the word “new”, and much less the word “schizophrenia”.

Delete “new” as I believe there are already other studies looking for what the Authors have been looking for. Instead of “new” I would use “possible new”. Instead of schizophrenia I would use “schizophrenia like psychosis”.

We used the word "new" twice in the context of the analyzed results. First in the title, then in the summary. Its subsequent uses refer to cited studies by other authors. In our opinion, the research approach we adopted in the research on schizophrenia presented here is unique due to its interdisciplinarity and the nature of the analyses conducted and deserves to be used in the description of our work.

We do not agree with the suggestion that the term schizophrenia should be treated interchangeably with the term "schizophrenia-like psychosis". In the literature on the subject, this term is used to describe psychoses with a clinical picture similar to schizophrenia, e.g. secondary to epilepsy or appearing late in life or accompanying dementia. Therefore, it is not an interchangeable term with schizophrenia.

AFFILIATIONS

I would not include English translations or instates. I would prefer to see those in original languages: Polish and Italian.

I would not include here any snail mail address.

I would not include all electronic addresses here. Please provide just one electronic address for one corresponding author.

Please agree to leave the authors' affiliation names in English. The presence of three different languages in the names of institutions may be confusing for readers, especially from countries using a non-Latin alphabet. We have removed snail mail addresses and electronic addresses, leaving the electronic address only for the corresponding author.

ABSTRACT

Never, but never, use any of these words to characterize schizophrenia: disease, illness, sickness.

In this work, in line with the authors' suggestion, we decided to remove the term mental illness or disease in relation to schizophrenia. These terms are still used in psychiatry to describe mental disorders of a particularly serious nature. They also appear frequently in scientific literature or are included in local and international legal acts. It is worth noting that the ICD 11 classification has its official name: International Statistical Classification of Diseases and Related Health Problems (ICD). However, we share the reviewer's opinion that both terms should cease to be used in the context of schizophrenia due to their potentially stigmatizing nature.

Please leave no acronym unexplained, eg describe what means “LC-ESI-MS/MS”.

Thank You for your comment, we have described the meaning of the indicated abbreviation and other missing abbreviations in the text. 

KEYWORDS

Please use no acronyms at this section, eg “MR”.

We replaced the acronym MR with its extension “magnetic resonance”.

Add “psychosis” to this list.

The term "psychosis" was introduced into the keywords of the text. It is worth noting, however, that the patients included in the study were at various stages of the disease, including those in remission of psychotic symptoms as a result of the treatment.

INTRODUCTION

Again, schizophrenia is not a “mental illness”. Schizophrenia is a syndrome, the greatest imitated syndrome in psychiatry, mimicked by many different conditions in medicine. Schizophrenia is not even a mental problem. It is indeed a psychiatric disorder, a syndrome that can be caused by many systemic, infectious, autoimmune, neurologic (etc.) conditions, eg “secondary schizophrenia”, “schizophrenia-like psychosis” or “pseudo-schizophrenia”. Only, when clinicians or researchers find no cause for psychosis they can use the concept of schizophrenia, primary schizophrenia, true schizophrenia or idiopathic schizophrenia. Please, read more in this subject here:

https://pubmed.ncbi.nlm.nih.gov/33099372/

https://pubmed.ncbi.nlm.nih.gov/34978980/

https://pubmed.ncbi.nlm.nih.gov/35523150/

https://pubmed.ncbi.nlm.nih.gov/32220147/

In this work, in line with the authors' suggestion, we decided to remove the term mental illness or disease in relation to schizophrenia. These terms are still used in psychiatry to describe mental disorders of a particularly serious nature. They also appear frequently in scientific literature or are included in local and international legal acts. It is worth noting that the ICD 11 classification has its official name: International Statistical Classification of Diseases and Related Health Problems (ICD). However, we share the reviewer's opinion that both terms should cease to be used in the context of schizophrenia due to their potentially stigmatizing nature.

We do not agree with the suggestion that the term schizophrenia should be treated interchangeably with the term "schizophrenia like psychosis". In the literature on the subject, this term is used to describe psychoses with a clinical picture similar to schizophrenia, e.g. secondary to epilepsy or appearing late in life or accompanying dementia. Therefore, it is not an interchangeable term with schizophrenia.

Whenever using “United States” the Authors should specify if they are writing about the “United States of America” or other countries “United States of Mexico”?

The term United States is commonly used to describe the United States of America in official documents as well. This second name also appears in most texts by American authors. However, we corrected the entry at the reviewer's request.

Once again, do not describe schizophrenia as a “disease”.

In this work, in line with the authors' suggestion, we decided to remove the term mental illness or disease in relation to schizophrenia. These terms are still used in psychiatry to describe mental disorders of a particularly serious nature. They also appear frequently in scientific literature or are included in local and international legal acts. It is worth noting that the ICD 11 classification has its official name: International Statistical Classification of Diseases and Related Health Problems (ICD). However, we share the reviewer's opinion that both terms should cease to be used in the context of schizophrenia due to their potentially stigmatizing nature.

The acronyms “IMS-MS”, “LC-ESI-MS/MS”, “WHO”, “MRI”, “MRS”, are not clearly explained. I mean, I find no correlation with words and letters.

Thank You  for your comment. We have explained all the abbreviations used in the text.

It is not only the “polygenic nature of schizophrenia” that troubles excellent studies like the Authors presents here. It is also the wrong concept that “schizophrenia is a genetic disease”. It is not. Schizophrenia is, indeed pseudogenetic syndrome. Please read more here:

https://pubmed.ncbi.nlm.nih.gov/31200193/

Thank You for drawing attention to the use of the word polygenic where we wanted to emphasize the complexity of the determinants and clinical picture of schizophrenia. There is no doubt that there are genes that increase the risk of developing schizophrenia, but as this and other work of our team shows, this risk is also associated with the occurrence of phenomena of an epigenetic nature relating to relational or social factors risk.

MATERIALS AND METHODS

I cannot understand why RESULTS and DISCUSSION came before METHODS in the article structure.

When preparing the work, we relied on the template provided to authors by the International Journal of Molecular Sciences, where immediately after the Introduction there are the Results, Discussion, and then Methods sections. In our work, we have changed the structure of the text according to the Reviewer's suggestion, the Methods section is located after the Introduction section.

All patients should have been submitted to 24 hour electroencephalography to exclude temporal lobe epilepsy, plus lumbar puncture to exclude encephalitis. The Authors should also have done this approach to all control subjects.

The study included people diagnosed with schizophrenia according to ICD-10 criteria, assessed by specialized and experienced psychiatrists from a leading clinical center. This means that 51 patients included in the research had a clinical picture and duration of symptoms typical for schizophrenia. In accordance with the current state of medical knowledge, etiological factors that could lead to a similar clinical picture were also excluded. According to diagnostic standards in psychiatry, not every patient diagnosed with schizophrenia undergoes many hours of EEG or lumbar puncture. These tests are performed in patients whose medical history indicates the need for in-depth neurological and differential diagnostics. The study group, analyzed in the reviewed text, includes patients with a well-established diagnosis - also through the analysis of the dynamics of the disease and response to treatment. People in the control group were healthy. In their case, the ethics committee would not agree to a lumbar puncture due to the potential health risks associated with such an examination. We added an appropriate caveat regarding the level of thoroughness of clinical diagnosis in the control group to the discussion of study limitations.

The Authors use too many self-report instruments, which fail a lot whenever used by patients with psychosis.

The Authors should never include patients with less than 18 years old. I would also never include patients with more than 65 years old. Adult age goes from18 to 65. Authors should be strict with this definition. Psychotic patients with 13 years old can later become schizoaffective, if a manic episode shows up later. It is very risky to contaminate a adult sample with teenagers or elderly. Another example: elderly cold contaminate sample with dementia. Schizophrenia is a syndrome of adulthood so it should be studied in adults. It is a completely different thing to make a study of psychosis among the teenagers or the elderly, as there are many other bias.

The study included patients up to 40 years of age. We do not agree with the argument that the inclusion of patients under 18 years of age is a study error. Since the development of the first diagnostic criteria for dementia praecox and then schizophrenia, it has been obvious that schizophrenia is a polyetiological disorder. Eugen Bleuler's observation that we are dealing here with a group of schizophrenias rather than a single disorder is still valid. It is worth remembering in this context that the diagnosis of dementia praecox was used for a long time in parallel with the new diagnosis of schizophrenia.

Schizophrenia is still a disorder diagnosed based on its clinical picture. Scientific research and clinical practice indicate that the same clinical picture of schizophrenia will be caused by various etiological factors or constellations of various etiological factors. Similar symptom complexes may have different causes. Hence, there is a special value in similarities between patients of different age, gender and other risk factors. The aim of our research is not to defend a monolithic concept of schizophrenia diagnosis, but to identify common aspects in risk factors and to define subtypes of the disorder.

In the differential diagnosis of schizophrenia, we strive to exclude other defined disorders that may be subject to treatment - e.g., autoimmune diseases. However, we are convinced that what the author of the review calls primary schizophrenia, true schizophrenia or idiopathic schizophrenia also have their risk factors, pathophysiological and pathological features. The group we are studying does not include misdiagnosed patients in whom a more careful clinician would have ruled out schizophrenia. These are patients diagnosed with schizophrenia in whom we can observe a number of subtle biological anomalies. There is an ongoing discussion on this matter in the literature on the subject. When discovering the determinants of some cases of schizophrenia, autistic spectrum disorders or depression, should these syndromes be excluded from syndromological psychiatric classifications? A different approach to this issue is observed in the USA and Europe.

We believe that the anomalies we describe in our study constitute a core aspect of schizophrenia. We hope that our work will contribute to the development of knowledge enabling the treatment of schizophrenia, not symptomatic as in the case of antipsychotic drugs, but causal treatment of some of its subtypes.

We share the author's doubts about the difficulties in diagnosing schizophrenia in teenagers. People presenting acute psychotic disorders for the first time in their lives may, over time, be diagnosed with schizophrenia, bipolar disorder, or schizoaffective disorder. Patients who experience a brief psychotic decompensation during the development of the Borderline personality may have a similar picture. Also in the latter case, we may be dealing with diagnostic errors. It sometimes happens that patients with post-traumatic disorders, especially those resulting in the development of complex dissociative disorders, are treated as having a diagnosis of schizophrenia. Young patients who hide addiction to psychoactive substances may present psychotic symptoms. Various somatic diseases may resemble schizophrenia. Experiments resulting from the search for one's identity, such as the creation of Tulpa, are also important. The authors of the text are aware of all these reservations and if they use the diagnosis of schizophrenia before the age of 18, they do so responsibly and after a thorough differential diagnosis and patient observation.

I would like to point out that although our text concerns largely biological phenomena, we do not reduce the etiology of schizophrenia to genetic conditions. The perspective expressed in this and the team's previous publications refers to risk factors including not only biological but also psychosocial factors, such as the multidirectional impact of early childhood trauma. This goal explains the multitude of diagnostic tools used in the study. They allow not only the analysis of the clinical picture but also the use of mechanisms for coping with difficult situations or experiencing trauma during childhood. The use of a larger number of tests was also related to the desire to analyze clinical material without ideological assumptions.

People using caffeine and/or tobacco should be studied in different groups, as these drugs have a lot of effects on symptoms and treatment.

Please provide chlorpromazine equivalent doses for all patients. It is always an interesting variable in studies involving psychotic and medicated patients.

Please, explain all acronyms, such as “K+”, “Na+”, “Mg2+”, “HOMA-IR”, “XN-9100”, “Corp.”, “PRO”, “PRO/e801”, “AG”, “BN II”, “Henkestr.”, “ACQUITY UPLC”, “MA”, “USA”, “NM20ZA”, “3T”, “T/m/s”, “3D”, “2D DWI”, “s/mm2” etc.

Thank You for your comment, we have described the meaning of the indicated abbreviations and other missing abbreviations in the text. 

Please do not provide address for companies such as “Henkestr. 127, 91052 Erlangen, Germany”

Following the Reviewer's suggestion, we have made appropriate corrections to the text.

I believe instead of “K+, Na+, Mg2+” it should be “K+, Na+, Mg2+”.

Thank You for your comment, we have made appropriate corrections to the text.

Please use Trade Mark (™) and Registered (®) symbols whenever adequate.

RESULTS

I cannot understand why RESULTS come before METHODS in the article structure.

When preparing the work, we relied on the template provided to authors by the International Journal of Molecular Sciences, where immediately after the Introduction there is the Results, Discussion and then Methods sections. In our work, we have changed the structure of the text according to the Reviewer's suggestion, the Methods section is located after the Introduction section.

I believe “ug/ml” is a misprint. Please use the Greek letter “µ” (mu), if you mean microgram.

 Thank You for your comment, we have made appropriate corrections to the text.

TABLES

Please explain following acronyms: “nM”, “ng/ml”, “mmol/l”, “nM”, “Mdn”, “Q1”, “Q3”, etc.

We have added abbreviation expansions in tables.

I believe “ug/ml” is a misprint. Please use the Greek letter “µ” (mu), if you mean microgram.

Thank You for your comment, we have made appropriate corrections to the text.

Table 3, Table 4  and Table 7 are way too long. Please make it shorter. Other options would be to produce a horizontal table instead of a vertical one. Or divide it in two smaller tables.

I would recommend moving Table 7 to the RESULTS section.

Following the Reviewer's recommendation, we moved Table 7 to the RESULTS section.

Table 8 has too much text. Please reduce it to the necessary minimum.  And explain all acronyms at the bottom.

Thank You for this comment. We have supplemented the list of abbreviations at the bottom of the table and also shortened the text in the table. 

FIGURES

I believe figures are out of place.

Images of laboratory machinery should be part of METHODS section, not INTRODUCTION.

Figure 1 at Page 4 is at very low quality, and I think they belong to the METHODS or RESULTS section, not INTRODUCTION.

We moved the chromatograph figure to the METHODS section.

Please explain the meaning of colors in Figure 2.

DISCUSSION

I am very disappointed to read this fragment “antipsychotic drugs (fluoxetine or S-citalopram).” Fluoxetine is an antidepressant drug, with no antipsychotic effect. I do not know what the Authors mean with “S-citalopram”. I know and use escitalopram and citalopram in my clinical practice for 15 years, but I am not aware of a drug called “S-citalopram”.

The drug name S-citalopram was taken from the original work from which it was quoted. We decided not to change it due to the fact that the essence of the quote does not concern the drug itself but a group of drugs. Additionally, the name S-citalopram is used less often - but it is chemically improved. It refers not to the marketing name of the substance but to its molecular structure. "Citalopram" available on the market is a racemic mixture of two stereoisomers (enantiomers): R-citalopram and S-citalopram (escitalopram) in a 50%/50% ratio. Escitalopram contains only the second stereoisomer.

Again, where I read “level of glutamate decreased after 10 days of therapy with both anti-psychotic drugs.” It should be antidepressant instead of “anti-psychotic”.

Thank you for your comment, we have made appropriate correction to the text.

Olanzapine is not a neuroleptic. It is instead a second generation antipsychotic with virtually no “extrapyramidal symptoms”. Haloperidol is a first generation antipsychotic, which is synonymous with neuroleptic. Beware: only first generation antipsychotics are neuroleptics. Second generation antipsychotics (eg olanzapine) and third generation antipsychotic (eg aripiprazole) are not neuroleptics.

We have corrected the issue related to the nomenclature of drugs used to treat schizophrenia in line with the reviewer's expectations. We would like to point out, however, that the term neuroleptics is sometimes used for both first- and second-generation antipsychotic drugs, and that the customary nomenclature in this respect differs between countries. However, we agree with the author of the review that the use of the term first- and second-generation antipsychotic drugs has its justification.

Thank you very much for pointing out the translation error. Of course, in the context of the above quote, it was about neurotropic drugs, not neuroleptics.

Limitations should be discussed here in a serious way. The authors should acknowledge that they did not make a proper exclusion of organic cause for psychosis in patients, nor brain anomaly in controls.

The review inspired us to modify the limitations of the work, which are included in the text.

Limitations

The study has a number of limitations. The group of patients included in the study is characterized by different age of onset, duration of psychosis and its stage. Both male and female participants were included in the study. This may affect the dispersion of the results. Schizophrenia diagnosis is made based on the clinical picture. This means that the groups of patients included in the study are potentially heterogeneous in nature. The differential diagnosis of patients included in the research was not uniform and It contained elements relating to the clinical picture. Therefore, patients were not subject to a uniform diagnostic procedure, including, for example, video-EEG examination and cerebrospinal fluid examination. However, none of the patients from the research and control groups showed features indicating the presence of anomalies in the MRI examination that would question the diagnosis of schizophrenia or indicate the need for further examination of the control group. The exclusion of mental and somatic disorders in the control group was based on patients' declarations and interviews, without a full medical examination or additional tests. Based on basic additional tests, no disorders requiring further diagnostics were found in any person from the control group.

Explain all acronyms such as “1H”, “ug/ml”, “G1-G16”, “TSH”, “ITQ”, “ICD-11”, “pH”, “H+”, “i.e.”, “BBB”, “mmol/l”.

Thank You for your comment, we have described the meaning of the indicated abbreviations and other missing abbreviations in the text. 

I believe it should be “H+” instead of “H+”, and “O2” instead of “O2”.

Following the Reviewer's suggestion, we have made appropriate corrections to the text. 

CONCLUSIONS

Authors should be more careful in this section. Information taken from “self-report” questionnaires can be very tricky to interpret in psychotic patients, even more in patients with schizophrenia. Anosognosia or lack of insight can be an important bias, whenever interpreting this kind of data.

REFERENCES

This section is way too long.

The publisher does not specify the limits on the number of citations.

Again I would recommend Authors to divide the article in two or even three smaller articles.

We agree with the author of the review that the text is long. However, the results we obtained would be difficult to comprehensively analyze in separate texts. The text presents a certain approach to clinical data, only a comprehensive analysis of which can allow for the assessment of the correctness of the conclusions drawn.

We are very grateful for all your comments and time, spent on our article!  We hope that the presented explanations and introduced corrections will be satisfactory to both the reviewers and You. Thank You again for your time and effort spent on improving our manuscript.

Round 2

Reviewer 2 Report

Comments and Suggestions for Authors

In my opinion "s-citalopram" should be written "escitalopram". Because "escitalopram" is the market name that we, physicians prescribe, and patients take.

I did not understand what was the point of citing the Tulpa phenomenon... but at least I learnt something esoteric for today!

Please do not bother me anymore with with review of this article.

Congratulations to the Authors.

Author Response

Dear Reviewer,

I would like to acknowledge the Reviewers for all their valuable comments and suggestions on our manuscript entitled: “Association of blood metabolomics biomarkers with brain metabolites and patient-reported outcomes as a new approach in individualized diagnosis of schizophrenia”. Reviewers’ remarks have made possible modifications to the manuscript towards, as we believe, a better presentation of the results. A list of recent changes introduced to the manuscript in response to specific issues raised by the Reviewer is given below. Actions taken are marked in the manuscript in “track changes” mode.

According to the Reviewer's suggestion we change the name "s-citalopram" into "escitalopram". This change in the manuscript is marked in green.

The mention of the Tulpa phenomenon was intended to highlight the complexity and diversity of factors that can influence the diagnosis of psychiatric disorders such as schizophrenia, especially in adolescents. Although esoteric and unusual, the Tulpa phenomenon was cited as an example of identity-seeking experiments that can sometimes be misinterpreted as psychotic symptoms. In the context of our discussion, Tulpa serves as an example of an atypical but significant aspect that professionals might consider during the diagnostic process. We wanted to emphasize the importance of a broad perspective and caution in the diagnostic process, keeping in mind that the unusual behaviors of young people can have various underlying causes, not necessarily related to schizophrenia.

Therefore, our goal was to signal how crucial it is for specialists to not only focus on apparent symptoms but also to consider less conventional aspects of behavior that may influence the diagnostic process. This is particularly important in the context of young patients, whose diversity of life experiences can affect the manifestations of their mental states.

In our study, we aimed to explore the full spectrum of schizophrenia's onset, including its early manifestations in adolescent. This approach allows us to gain a comprehensive understanding of the disorder, contributing to better diagnostic criteria and treatment strategies that are age-appropriate and effective for all age groups.

We employed rigorous criteria for the diagnosis of schizophrenia in younger participants, ensuring that our methods were aligned with the best practices in psychiatric evaluation. We took extra precautions to differentiate between schizophrenia and other developmental disorders that can present similar symptoms in this age group. Our diagnostic process involved a thorough assessment, including detailed clinical interviews, family history analysis, and when necessary, longitudinal follow-up to confirm the diagnosis. By including younger individuals in our study, we aimed to provide a more inclusive and detailed picture of schizophrenia, which is essential for developing age-appropriate intervention strategies and enhancing our understanding of this complex disorder across different stages of life.

Dear Academic Editor,

we analysed all the Reviewers' comments. We responded to all comments by making appropriate changes to the manuscript. After that, Reviewer 2 had no further comments except a few minor ones, which we also responded to. If the Academic Editor has any specific expectations apart from the Reviewers' comments, please indicate the specific comments that we should introduce to the manuscript. This will make it easier for us to work with the manuscript if there is anything else required.

Thank You for Your consideration of this manuscript.

Sincerely,

Wirginia Krzyściak